# AITTI: LEARNING ADAPTIVE INCLUSIVE TOKEN FOR TEXT-TO-IMAGE GENERATION

## ABSTRACT

Despite the high-quality results of text-to-image generation, stereotypical biases have been spotted in their generated contents, compromising the fairness of generative models. In this work, we propose to learn *adaptive inclusive tokens* to shift the attribute distribution of the final generative outputs. Unlike existing de-biasing approaches, our method requires neither explicit attribute specification nor prior knowledge of the bias distribution. Specifically, the core of our method is a lightweight *adaptive mapping network*, which can customize the inclusive tokens for the concepts to be de-biased, making the tokens *generalizable to unseen concepts* regardless of their original bias distributions. This is achieved by tuning the adaptive mapping network with a handful of balanced and inclusive samples using an anchor loss. Experimental results demonstrate that our method outperforms previous bias mitigation methods without attribute specification while preserving the alignment between generative results and text descriptions. Moreover, our method achieves comparable performance to models that require specific attributes or editing directions for generation. Extensive experiments showcase the effectiveness of our adaptive inclusive tokens in mitigating stereotypical bias in text-to-image generation. The code will be publicly available.

## 1 INTRODUCTION

Text-to-image (T2I) generation has gained widespread usage thanks to its ability to produce visual content from user-specified text descriptions. However, alongside the technical achievements, concerns have arisen regarding the presence of stereotypical biases in the generated outputs, as analyzed by various studies (Ghosh & Caliskan, 2023; Chinchure et al., 2023; Wang et al., 2023; Bianchi et al., 2023; Wang et al., 2024; Jha et al., 2024).

The unfairness in the T2I generation is usually reflected by an unequal representation of different social groups. It is observed that in circumstances where no specific attributes are specified in a human-related prompt, the T2I models tend to generate human figures resembling certain genders and races, reinforcing harmful social discrimination (Ghosh & Caliskan, 2023; Bianchi et al., 2023). In particular, some occupations are strongly associated with specific genders following stereotypes, *e.g.*, male for doctor and female for nurse. Furthermore, negative concepts like poverty or unattractiveness lead to the generation of people from colored races (Bianchi et al., 2023; Ghosh & Caliskan, 2023; Jha et al., 2024). Exposure to such an unfair environment will strengthen the stereotypes and biases, justify hate in our society, and undermine the rights of minority groups to be treated fairly. While unfair training datasets are often recognized as the origins of biases in large models, the pervasive nature of stereotypes in our society makes the complete elimination of biases challenging. This triggers the necessity to design de-biasing algorithms to mitigate biases in the pre-trained models.

The main goal of this paper is to devise a bias mitigation method for biased concepts[1]. An ideal inclusive T2I model yields results with evenly distributed sensitive attributes across all attribute classes when all classes are factually correct and no attribute-related instructions are provided. A crucial aspect of a fair model is its ability to generate inclusive outcomes without direct instruction

---

[1]For clarity, we define the key terms as: *sensitive attributes* $\mathcal{A}$ refer to the attributes of interest for achieving fairness (*e.g.*, gender), and *attribute classes* $\mathcal{A}_c$ denotes possible classes of the attribute (*e.g.*, female and male). *Biased concepts* $\mathcal{C}$ describe the concepts that contain potential stereotypes (*e.g.*, doctor).

regarding the target attribute class. Besides, users' unawareness of potential biases related to a target concept should be respected. Therefore, we argue that, given a neutral concept, a good de-biasing algorithm should (1) achieve fairer results without explicit specification of the target attribute class during generation, and (2) require no prior knowledge of the original bias distribution associated with the concept (*e.g.*, the doctor concept is stereotypically biased towards males). In practice, by specifying the sensitive attribute of interest rather than specific attribute classes in the provided prompt (*e.g.*, "A <gender-inclusive> doctor" instead of "A female doctor"), the users can obtain fairer generative results regarding the attribute they care.

Achieving the aforementioned inclusive properties is non-trivial in the absence of direct attribute specification and prior knowledge about the bias distribution. Our study shows that the simple approach of learning a fixed inclusive token via Textual Inversion (TI) (Gal et al., 2022) fails in mitigating biases in concepts originally biased toward different classes (see Sec. 3.1 for detailed discussion). In addition, it also risks modifying the semantic concepts specified in the original prompt. In this work, we follow the mainstream prompt tuning method but aim at learning **adaptive inclusive tokens** that do not specify any target class, and yet can shift the bias attribute in generation outcomes towards a more equitable distribution, regardless of the class it was originally biased to.

We hypothesize that the token embeddings of a biased concept, *e.g.*, the doctor concept, encode information regarding its bias distribution. Hence, to enable the adaptability of the learned inclusive tokens across various biased concepts, a lightweight **adaptive mapping network** can be used to find the optimal inclusive token, taking into account the target concept. In addition, an **anchor loss** is proposed to guide the desired properties of the inclusive token, minimizing the discrepancy between de-noising UNet predictions of prompt with inclusive token and prompt with the ground truth attribute class. The anchor loss ensures the inclusive token has an aligned impact with attribute class tokens and alternates among all possible classes.

Our proposed method is validated on the widely adopted Stable Diffusion (SD) framework (Rombach et al., 2022). Utilizing small-scale balanced datasets generated by the SD model itself to train our adaptive inclusive tokens, the fairness of the final model outputs is significantly improved. Notably, the learned adaptive inclusive tokens demonstrate generalizability to unseen concepts and prompts, and can be concatenated to mitigate multiple biases along various attributes.

The contributions of our work are as follows: We introduce a simple yet effective prompt-tuning approach to achieve inclusive text-to-image generation without attribute specification or prior knowledge of biased concepts. In particular, we propose the adaptive mapping network together with the anchor loss to address the issue of generalizability across different attribute class dominations. Extensive experiments show the effectiveness of our method both quantitatively and qualitatively.

## 2 RELATED WORK

### 2.1 BIAS IN TEXT-TO-IMAGE GENERATION

Comprehensive analyses have been conducted to study the bias and unfairness observed in T2I generation (Bianchi et al., 2023; Chinchure et al., 2023; Ghosh & Caliskan, 2023; Jha et al., 2024; Wang et al., 2023; 2024). Wang et al. (2023) first introduce the Implicit Association Test (IAT) (Greenwald et al., 1998) from social psychology to measure biases in the task of T2I. IAT is designed to reveal implicit biases that an individual may hold unconsciously towards certain concepts. By experimenting with the valence and stereotype IATs on T2I output images, it is found that valence and stereotypical biases exist in state-of-the-art T2I models at various scales, *e.g.*, the pleasant attitude is significantly biased towards straight sexual orientation than homosexual ones. Bianchi et al. (2023) study a wide range of stereotypes related to gender, race, nationality, and other identities associated with traits, occupation, and even object descriptions. They conclude that attempts to either specify counter-stereotype prompts by users or add system "guardrails" cannot prevent stereotypes from spreading in the T2I results. Ghosh & Caliskan (2023) spot the over-representation of Caucasian males in general terms like "person" and the over-sexualization of women of color without specification. TIBET (Chinchure et al., 2023) proposes to identify and measure biases in any T2I models using counterfactual reasoning, which breaks the limitation of pre-defined bias axes in previous studies. Wang et al. (2024) propose BiasPainter which provides an automatic and systematic study of gender, race, and age biases. It augments a seed image of a clear identity with queries of profes-

sions, activities, types of objects, and personality traits and compares the attributes of the identity between augmented and seed images to identify biases in queries. Jha et al. (2024) focus on exploring geo-cultural stereotypes in T2I models with a large scale of nationality-based identity groups. It is revealed that the severity of bias varies for different identity groups and the "default" (without specification on bias attributes) representations of identity groups contain stereotypical appearances.

## 2.2 BIAS MITIGATION IN TEXT-TO-IMAGE GENERATION

Various approaches have been developed to alleviate stereotypical biases in T2I generation, including model fine-tuning (Runway, 2023; Kim et al., 2023; Shen et al., 2024), prompt tuning (Gal et al., 2022; Bansal et al., 2022; Zhang et al., 2023; Li et al., 2023), concept editing (Orgad et al., 2023; Gandikota et al., 2024), and inference guidance (Friedrich et al., 2023; Parihar et al., 2024). A straightforward strategy is to fine-tune the entire T2I model on a large-scale dataset that is carefully synthesized to cover various classes of bias attributes (Runway, 2023). Besides, techniques of fine-tuning the sampling process of the diffusion model have been proposed so that distributional constraints on the generative outputs, which is the direct interpretation of fairness, can be applied (Kim et al., 2023; Shen et al., 2024). However, such fine-tuning methods require a heavy load of computations. Prompt tuning-based methods aim to modify or add textual tokens to affect the T2I generation outputs. Gal et al. (2022) learn a fairer word for a biased concept from a small dataset. Bansal et al. (2022) explore the impact of ethical interventions added to the original prompts on the fairness of generative results. Li et al. (2023) introduce a fair mapping network that projects the text embeddings of a neutral prompt to the middle of prompts with all possible classes. The above prompt tuning methods do not modify the prompt to provide explicit guidance during generation. On the contrary, Zhang et al. (2023) propose to overfit one class token for each attribute class by image prompts and apply ad-hoc post-processing to loop over all combinations of target attributes to achieve inclusive generation. Orgad et al. (2023) and Gandikota et al. (2024) apply model editing to enforce the generation of non-stereotypical classes by optimizing the cross-attention weights of the diffusion models. However, determining the appropriate editing strengths is challenging considering the large variations in bias strengths across different concepts and classes. Another line of retraining-free approaches involves incorporating desired guidance during inference. Friedrich et al. (2023) apply fair guidance at inference by steering biased concepts in predefined directions to enhance fairness. While this method avoids the need for training or fine-tuning, it relies on prior knowledge of the biased concept and requires each generation to be guided by a specific semantic direction. Similarly, Parihar et al. (2024) employ a pre-trained h-space classifier to provide explicit distribution guidance during inference. Although this method does not require retraining the diffusion model, it demands additional effort to train the classifier and increases the computational cost of inference due to classifier-guided optimization. Our approach adopts the prompt tuning approach without necessitating heavy model fine-tuning or prior knowledge about the original bias distribution, making it computationally efficient and highly generalizable.

## 2.3 PROMPT TUNING

Prompt tuning is a technique that adapts a large language model (LLM) to new concepts by optimizing some prompt parameters with the model weights fixed. Prompt tuning has been applied to various downstream tasks such as image classification (Zhou et al., 2022b;a), customized generation (Gal et al., 2022; Ruiz et al., 2023), and bias mitigation (Gal et al., 2022; Kim et al., 2023). Zhou *et al.* (Zhou et al., 2022b;a) propose to optimize trainable context tokens in class prompts to boost the performance of zero-shot classification using CLIP classifier (Radford et al., 2021) and further tailor the tokens to be input-conditioned for better generalization to unseen classes. Textual Inversion (Gal et al., 2022) is proposed to invert a visual concept from a small set of images to a new pseudo word and customize generation on the visual concept. Similarly, Ruiz et al. (2023) and Kumari et al. (2023) update a trainable token together with some parts of the model to represent a particular visual content. The ability of prompt tuning to mitigate biases has been demonstrated by Gal et al. (2022) as mentioned in the previous subsection. Kim et al. (2023) adopt a similar prompt tuning approach. However, they optimize the tokens on the sampling stage of the diffusion model, leading to an extensive amount of computation and memory required. Our method tackles the limitation of fixed inclusive tokens on the transferability to different domination classes, as observed in both previous methods, by learning adaptive inclusive tokens.

## 3 METHODOLOGY

### 3.1 PRELIMINARIES

**Diffusion Model.** Stable Diffusion (Rombach et al., 2022) is a commonly used latent diffusion model for image generation. With classifier-free guidance of textual conditions, SD demonstrates excellent performance in the T2I generation task. During each training step, a training image $x_0$ is first encoded into a latent space as $z_0$ by a pre-trained image encoder $\mathcal{E}(\cdot)$. Then, a noise latent and a de-noising timestep $t$ are sampled to compute the ground truth noise $\epsilon$, which will be added to the encoded image to obtain the noisy latent $z_t$ in the current timestep. On the other side, a textual prompt $T$ that describes the image is tokenized to token embeddings $v_T$ and encoded to textual embeddings $e_T$ by the pre-trained CLIP text encoder (Radford et al., 2021). Subsequently, a de-noising UNet $\epsilon_\theta$ (Ronneberger et al., 2015) takes in the noisy image latent, the de-noising timestep, and the textual condition to predict the noise added to the image latent. The learning objective of the de-noising stage is formulated as follows:

$$\mathcal{L}_{denoise} = \mathbb{E}_{z_0, \epsilon \sim \mathcal{N}(0,1), t, e_T} \left[ \| \epsilon - \epsilon_\theta \left( z_t, t, e_T \right) \|_2^2 \right] . \tag{1}$$

During inference, the de-noising UNet predicts the noise to be removed at each timestep from a Gaussian noise latent, conditioned on textual input.

**Is Textual Inversion Effective for De-biasing?** Our method is inspired by the Textual Inversion framework introduced by Gal et al. (2022). Their paper briefly discussed the application of the Textual Inversion technique on bias reduction and provided a simple demonstration. Specifically, they curate a small and diverse dataset for a particular concept and learn a fairer word to substitute the original concept. The effectiveness of Textual Inversion in bias mitigation demonstrates that the fairness of T2I results can be significantly affected by the text condition, and it is feasible to invert a more balanced distribution to pseudo words from a small set of images. To enable the generalizability to unseen concepts, we revise their approach by disentangling the concept information from the learned tokens, making them solely represent the inclusiveness of a biased attribute. To elaborate, while the original Textual Inversion paper proposes learning a pseudo word <inclusive-doctor> to replace the original word "doctor", the revised variant learns a pseudo word <gender-inclusive> applicable to any human-related concepts for a fairer generation in gender. The experimental results of this naïve revised version of the Textual Inversion approach (labeled as TI), are reported in Tab. 1.

Although the approach effectively reduces bias in gender and race attributes numerically, it exhibits two notable limitations in its generated outputs. Firstly, a single learnable inclusive token is inadequate in mitigating biases in concepts originally biased towards different classes. Secondly, the semantic meanings of certain concepts have been altered by the learned inclusive token. Upon testing with unseen occupations that are stereotypically associated with different genders (*e.g.*, "software developer" and "flight attendant" are stereotypically dominated by male and female figures,

| Method | MD | FD | Imbalance |
|--------|--------|--------|-----------|
| SD1.5 | 0.5166 | 0.4401 | 0.0765 |
| TI | 0.4147 | 0.2720 | 0.1427 |
| Ours | **0.1436** | **0.1216** | **0.0220** |

(a) Quantitative comparison of the distribution discrepancy $\mathcal{D}_{KL} \downarrow$ across occupations stereotypically dominated by different genders. Each domination group consists of 5 unseen occupations (detailed lists are provided in the supplementary materials). **MD**: male-dominated occupations. **FD**: female-dominated occupations.

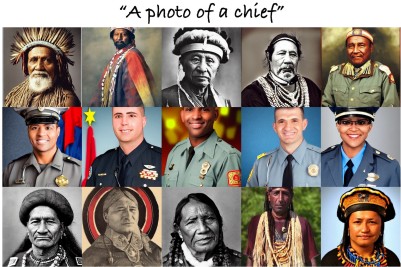

"A photo of a chief"

(b) Visual results. All images are generated with the same random seed. The caption above indicates the base prompt $T(c)$. **Top**: Rombach et al. (2022); **Middle**: revised Gal et al. (2022); **Bottom**: ours.

Figure 1: Limitations of revised Textual Inversion method with fixed inclusive token for de-biasing. (a) Imbalanced results across different bias distributions. (b) semantic drifting of visual concepts.

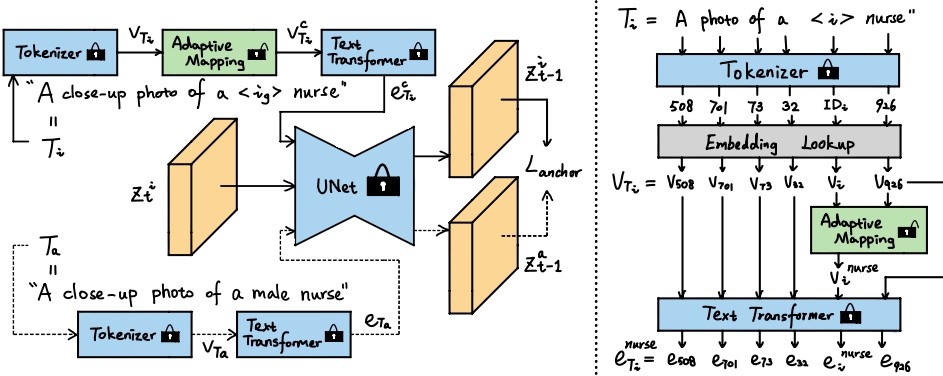

Figure 2: Framework of our proposed adaptive inclusive token for text-to-image generation. The blue color indicates frozen weights, and the green color indicates trainable weights. **Left**: single training stage. **Right**: details of text model with adaptive mapping network. The adaptive inclusive token is concept-specific. $TokenIDs$ are for illustration only.

respectively), results in Fig. 1a indicate that the revised Textual Inversion method significantly enhances fairness in occupations dominated by females. However, it shows less effectiveness in male-dominated occupations when using a single inclusive token. This leads to an increased imbalance in distribution discrepancy across different gender-dominated groups. This observation aligns with the findings by Kim et al. (2023), who conclude that the transferability of their de-stereotyping prompt is limited to unseen classes sharing the same dominant attribute class. Such sub-optimal results suggest that the token learns to simply generate more male figures instead of understanding the fair distribution. In addition, as illustrated in Fig. 1b, the term "chief" appears to deviate from its original semantic in the SD model under the influence of the inclusive token learned via the revised Textual Inversion method. Though the drifted representations of "chief" figures are not erroneous, given the term's broad definition, our objective is to ensure the learned inclusive tokens exclusively affect the biased attributes of the generated human figures without altering the semantics of the biased concepts.

## 3.2 ADAPTIVE INCLUSIVE TOKEN

To address the aforementioned limitations, instead of employing a fixed inclusive token for *all* concepts, we introduce a lightweight adaptive mapping network to predict the inclusive token tailored for each specific concept. To prevent the learned token from capturing irrelevant information from the training set that potentially causes semantic drift on bias concepts and degrades fairness in the generated results, an anchor loss is proposed to constrain the effect of the token. The framework of our method is illustrated in Fig. 2. Details of our proposed modules are described below.

**Adaptive Mapping Network.** As shown in the right side of Fig. 2, the adaptive mapping network $\mathcal{F}_{am}$ is introduced after the token embedding lookup and before the text transformer in our framework. The inclusive prompt $T_i$ contains an inclusive token denoted by a placeholder $$ $\in \{<i_g>,<i_r>,<i_a>\}$, representing <gender-inclusive>,<race-inclusive>,<age-inclusive>, respectively. The placeholder is processed by the text tokenizer and the token embedding lookup table along with other words, resulting in an initial inclusive token embedding $v_i$. Then, the inclusive token embedding $v_i$ and the biased concept embeddings $v_c$ are jointly fed into the adaptive mapping network to predict a concept-adaptive inclusive token $v_i^c$. The fundamental hypothesis is that the token embeddings of a concept inherently contain necessary information about its bias distribution. Therefore, customizing the inclusive tokens for each concept facilitates the selection of a more suitable token to effectively shift the final distribution towards fairness regardless of its original distribution. Finally, the adaptive inclusive token $v_i^c$ substitutes $v_i$ in the original token embeddings $v_{T_i}$ to form $v_{T_i}^c$, which is then forwarded to the text transformer model. The overall process of the text model with adaptive mapping is outlined in Algorithm 1.

**Anchor Loss.** Given the issue of concept drifting in the revised Textual Inversion results, as discussed in Sec. 3.1, we hypothesize that the issue may stem from ambiguous instructions on what information to be captured by the inclusive token during training. To guide the learnable token

---

**Algorithm 1** Text Model with Adaptive Mapping Network

---

1: **Input:** Tokenizer $\mathcal{F}_{tokenizer}$, Token lookup table $\mathcal{L}$, Text transformer $\mathcal{F}_{text}$
2: **Input:** Inclusive prompt $T_i = [T(c); ]$, where $T$ is the base prompt with biased concept $c$ and $ \in \{<i_g>, <i_r>, <i_a>\}$ is the inclusive placeholder
3: **Input:** Adaptive Mapping Network $\mathcal{F}_{am}$
4: $TokenIDs = \mathcal{F}_{tokenizer}(T_i)$
5: Token embeddings: $v_{T_i} = \mathcal{L}(TokenIDs) = [v_T; v_i]$
6: Concept-adaptive inclusive token embedding: $v_i^c = \mathcal{F}_{am}(v_i, v_c)$
7: Update $v_{T_i}$ to $v_{T_i}^c = [v_T; v_i^c]$
8: Text embedding: $e_{T_i}^c = \mathcal{F}_{text}(v_{T_i}^c)$
9: **Return** $e_{T_i}^c$

---

on the desired property for it to learn and promote fairness, we add constraints to the UNet noise prediction with inclusive text condition, as it represents the influence of text condition on the final generative outputs. Previously, Li et al. (2023) interpret fairness constraints as requiring the text embeddings of a de-biased prompt to be of equal distances from those of all class-specific prompts for one bias attribute. However, we argue that the notion of fairness and inclusiveness should not entail an average distance across all possible classes, but involve an equal-possibility shifting within those classes. Therefore, we proposed an anchor loss that is formulated as follows:

$$\mathcal{L}_{anchor} = \mathbb{E}_{z_0, \epsilon \sim \mathcal{N}(0,1), t, e_{T_i}^c, e_{T_a}} \left[ \left\| \epsilon_\theta \left( z_t, t, e_{T_i}^c \right) - \epsilon_\theta \left( z_t, t, e_{T_a} \right) \right\|_2^2 \right]. \tag{2}$$

The common de-noising process notations follow Eqn. (1). Here, $e_{T_i}^c$ indicates the text embeddings of inclusive prompt $T_i$, and $e_{T_a}$ represents the text embeddings of anchor prompt $T_a = [T(c); a]$ where the inclusive token is replaced by the ground truth attribute class $a$ of the training sample. For example, for a training sample of a female firefighter with gender as the target bias attribute, if $T_i$ is "a photo of a $<i_g>$ firefighter" then $T_a$ will be "a photo of a female firefighter". Rewarding the similarity of UNet noise prediction under $e_{T_i}^c$ and $e_{T_a}$ conditions has two potential benefits. Firstly, it provides indications that the learned inclusive token should specifically affect certain attributes of the output, *e.g.*, gender, as the anchor words. Secondly, it facilitates inclusive generations by allowing the effect of the inclusive token to shift among all possible classes.

The overall training objective for our adaptive inclusive token is defined as:

$$\mathcal{L} = \mathcal{L}_{denoise} + \lambda \cdot \mathcal{L}_{anchor}, \tag{3}$$

where $\mathcal{L}_{denoise}$ is calculated based on the inclusive prompt $T_i$, and $\lambda$ is the weighting parameter.

## 4 EXPERIMENTS

### 4.1 EXPERIMENTAL SETUP

**Scope.** We validate our adaptive inclusive token in mitigating three commonly studied bias attributes - gender, race, and age: $\mathcal{A} \in \{gender, race, age\}$. These attributes reflect harmful sexism, racism, and ageism observed in real life. Regarding gender bias, we consider binary classes of male and female, acknowledging its limitation in representing non-binary genders. However, we argue that identifying certain appearances as non-binary genders may reinforce stereotypes within these less-represented social groups, and therefore should be avoided until more carefully collected public data on non-binary genders is available. For racial bias, we refer to the seven racial groups from the FairFace dataset (Karkkainen & Joo, 2021). To simplify the analysis, we group East Asian and Southeast Asian into a single category Asian, resulting in six distinct groups: White, Black, Asian, Middle Eastern, Indian, and Latino Hispanic. As for age bias, binary groups of young and old people are examined. In this study, we focus on neutral bias concepts related to human figures, particularly in occupations: $\mathcal{C} \in \{neutral\ occupations^2\}$.

---

[2]Neutral occupations refer to those that are factually correct with all attribute classes. *e.g.*, "waitress" is not a neutral occupation since it infers female figures in its definition.

Table 1: Comparisons with baseline methods on fairness, quality, and text alignment of generative results across three bias attributes (See evaluation metrics details in Sec. 4.1). Abbreviations are used for simplicity: **SD1.5**: Rombach et al. (2022). **ITI-GEN**: Zhang et al. (2023). **TIME**: Orgad et al. (2023). **FD**: Friedrich et al. (2023). **EI**: Bansal et al. (2022). **FM**: Li et al. (2023). **TI**: Gal et al. (2022). * indicates editing-based methods that require careful tuning of editing strengths to achieve reasonable results. The best fairness results in each category are marked by **bold**.

| Methods | Gender | | | Race | | | Age | | |
|---|---|---|---|---|---|---|---|---|---|
| | $\mathcal{D}_{\mathcal{KL}}\downarrow$ | FID$\downarrow$ | CLIP$\uparrow$ | $\mathcal{D}_{\mathcal{KL}}\downarrow$ | FID$\downarrow$ | CLIP$\uparrow$ | $\mathcal{D}_{\mathcal{KL}}\downarrow$ | FID$\downarrow$ | CLIP$\uparrow$ |
| SD1.5 | 0.3584 | 281.12 | 0.2823 | 0.5973 | 281.12 | 0.2823 | 0.2319 | 281.12 | 0.2823 |
| With attribute specification or prior knowledge on the bias distribution | | | | | | | | | |
| ITI-GEN | **0.0078** | 278.21 | 0.2753 | **0.3699** | 247.05 | 0.2679 | **0.1560** | 243.09 | 0.2648 |
| TIME* | 0.2908 | 277.79 | 0.2733 | 0.5463 | 270.03 | 0.2663 | 0.2285 | 271.09 | 0.2738 |
| FD* | 0.2420 | 278.10 | 0.2718 | 0.4987 | 277.64 | 0.2738 | 0.2246 | 280.33 | 0.2740 |
| Without attribute specification | | | | | | | | | |
| EI | 0.1666 | 283.52 | 0.2758 | 0.6033 | 281.11 | 0.2745 | 0.2258 | 289.82 | 0.2773 |
| FM | **0.1174** | 222.82 | 0.2341 | 0.3722 | 220.37 | 0.2391 | 0.3823 | 255.72 | 0.2402 |
| TI | 0.2590 | 283.38 | 0.2777 | 0.8065 | 275.32 | 0.2799 | 0.3113 | 286.22 | 0.2823 |
| Ours | 0.1298 | 272.35 | 0.2789 | **0.3625** | 277.15 | 0.2808 | **0.2168** | 268.53 | 0.2798 |

**Training Data.** We use 24 occupations that span diverse bias attributes as our training occupations. For each occupation and attribute class, 20 images are generated using SD1.5[3] (Rombach et al., 2022) with 50 de-noising steps. The RetinaFace detector (Deng et al., 2020) is applied to filter generation without valid faces. In total, there are $(24 \times 2 \times 20)$ images for gender attribute, $(24 \times 6 \times 20)$ for race, and $(24 \times 2 \times 20)$ for age. Details on preparing training data can be found in the supplementary materials.

**Implementation Details.** Our main experiments are conducted on the SD1.5 network, yet the generalizability to other models is demonstrated in Appendix Sec. E.7. By default, the inclusive token $v_i$ is initialized to the embedding of the token "individual", which we find to be a natural inclusive token (detailed discussion on natural inclusive tokens can be found in the supplementary materials). The length of our adaptive inclusive token is set to 1, resulting $v_i^c \in \mathbb{R}^{1 \times 768}$, as 768 is the token embedding dimension of the SD1.5 text model. The models are trained for 3000 steps with batch size = 1 (gradient accumulation steps = 4), taking approximately 1 hour on one NVIDIA A100 GPU. The base learning rate is set to $5 \times 10^{-4}$ and we employ the $AdamW$ optimizer. As for training prompts, we follow Textual Inversion (Gal et al., 2022) to use the ImageNet templates describing objects, but add adjectives to describe the attribute of interest. The template training prompts can be found in the supplementary materials. For the adaptive mapping network, we use transformer architecture with six attention heads and four transformer blocks.

**Evaluation Protocols.** To evaluate the performance of mitigating stereotypical biases in T2I generations, we construct a test set of unseen 24 occupations excluded from the training set. For each biased concept, we analyze 100 images with valid faces to measure the distributions of different attributes. All images for evaluation are generated using the test prompt "a photo of a $$ {occupation}" with 25 de-noising steps. The racial inclusive token $<i_r>$ is introduced after 10 sampling steps on base prompt $T(c)$ to better balance six classes. Attribute classifiers are employed to detect and gather statistics on the distributions among various attribute classes in the generated images. Following previous works (Zhang et al., 2023; Gandikota et al., 2024; Kim et al., 2023), we utilize the CLIP (Radford et al., 2021) zero-shot classifier to classify sensitive attributes, prompting with "a photo of a [attribute] person".

**Evaluation Metrics.** Three metrics are used for evaluation. Firstly, following Zhang et al. (2023), the fairness of de-biased attribute distribution is assessed by the KL divergence ($\mathcal{D}_{\mathcal{KL}}$)$\downarrow$ against an even distribution. Secondly, Fréchet Inception Distance (FID)$\downarrow$ is used to quantify the image quality.

---

[3]runwayml/stable-diffusion-v1-5

"A photo of a detective"                "A photo of a doctor"

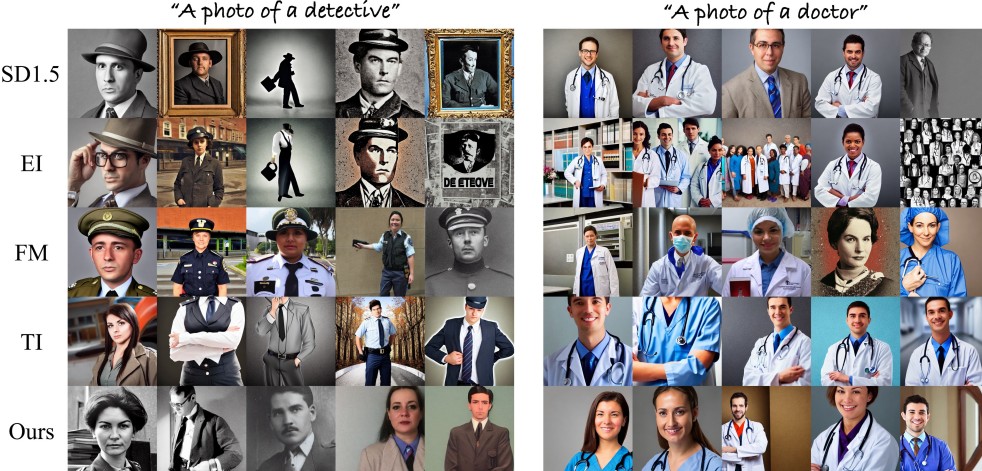

(a) Visual evaluation on stereotypically male-dominated occupations.

"A photo of a fashion designer"        "A photo of a yoga instructor"

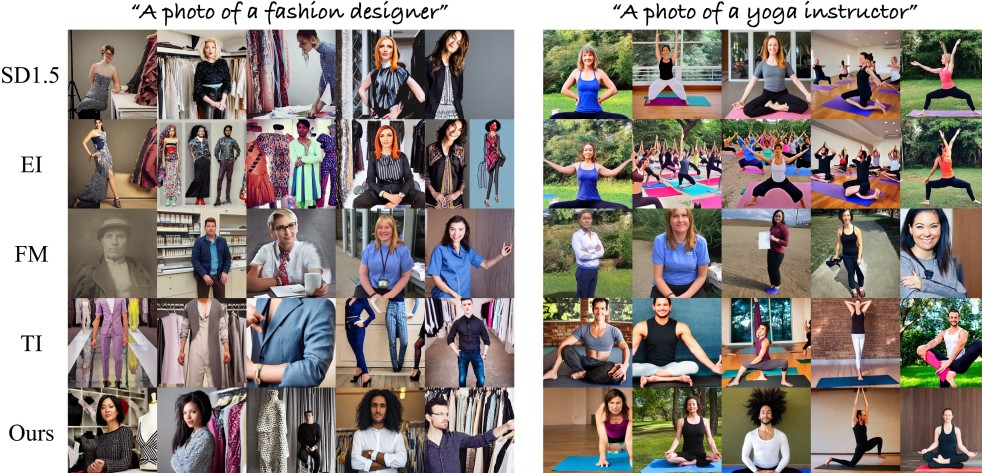

(b) Visual evaluation on stereotypically female-dominated occupations.

Figure 3: Qualitative evaluation on gender bias mitigation. All images are generated with the same random seed. The captions above indicate the base prompt $T(c)$.

Table 2: Ablation studies on proposed components for single-bias mitigation.

| $L_{anchor}$ | $\mathcal{F}_{am}$ | Gender $\mathcal{D}_{\mathcal{KL}} \downarrow$ | Race $\mathcal{D}_{\mathcal{KL}} \downarrow$ | Age $\mathcal{D}_{\mathcal{KL}} \downarrow$ |
|---|---|---|---|---|
|  |  | 0.2590 | 0.8065 | 0.3113 |
| ✓ |  | 0.1822 | **0.3540** | 0.3999 |
|  | ✓ | 0.3981 | 0.8090 | 0.2581 |
| ✓ | ✓ | **0.1298** | 0.3625 | **0.2168** |

Lastly, to evaluate the extent of concept drifting, we report the alignment between generated images and prompts using the CLIP-Score (CLIP)↑.

## 4.2 EXPERIMENTAL RESULTS

**Single Bias.** The quantitative comparisons of our proposed method against baselines in mitigating single bias are shown in Tab. 1. The implementation details of baseline methods can be found in appendix Sec. C. The methods are categorized into two groups: those with attribute specification or editing direction guidance and those without. As shown in the table, the methods with attribute specification generally yield good inclusive results. However, as mentioned in Sec. 1, we believe

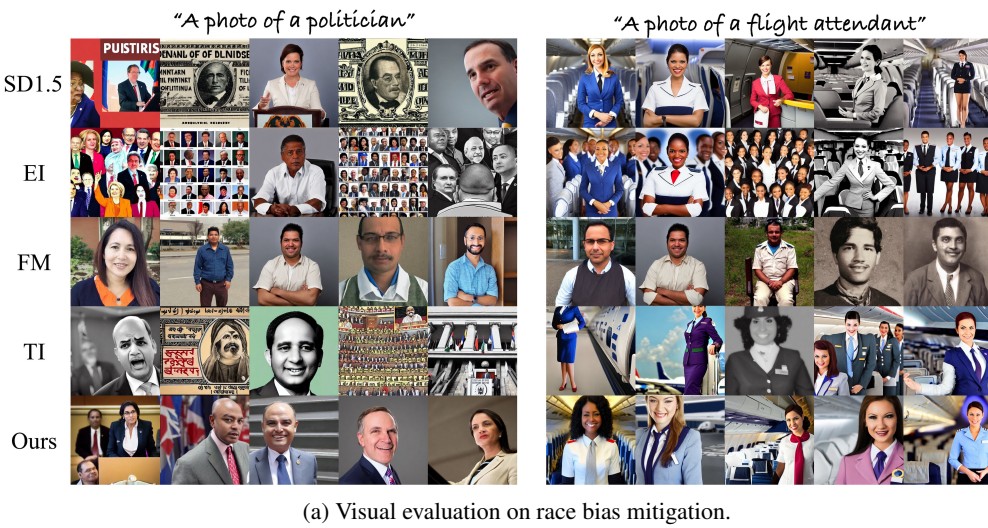

(a) Visual evaluation on race bias mitigation.

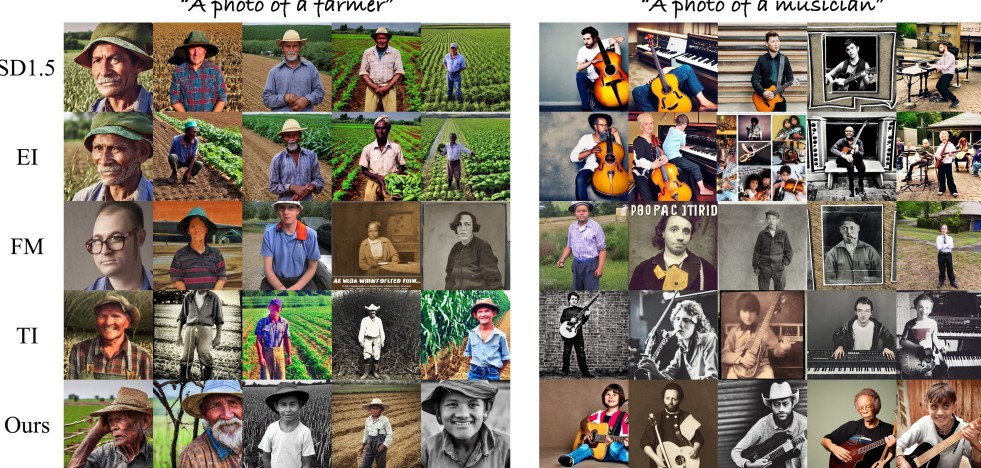

(b) Visual evaluation on age bias mitigation.

Figure 4: Qualitative evaluation on race and age biases mitigation. All images are generated with the same random seed. The captions above indicate the base prompt $T(c)$.

achieving fairer results without specifying attribute classes or editing direction poses a more meaningful and challenging task. Our approach excels in mitigating race and age biases without requiring additional information and is comparable to the best performance in gender bias mitigation in terms of numerical metrics. Qualitative comparisons of approaches that do not rely on additional information are shown in Fig. 3 and Fig. 4. Our method promotes better inclusiveness while maintaining the semantic meaning of a given concept. In addition, our method is model-agnostic and generalizes to any T2I models as long as the text encoder can handle newly learned embeddings. Our results on the SD2.1 (Rombach et al., 2022) and SDXL (Podell et al., 2024) models are shown in appendix Sec. E.7.

Despite the lower distribution discrepancy of Li et al. (2023) in gender bias (FM in Table 1), its CLIP-Score suffers a considerable decline, indicating a lack of visual semantics related to the target occupation in its generated images. Such severe semantic drifting from input prompts is evident in the qualitative evaluation (Fig. 3 and Fig. 4). Additionally, Bansal et al. (2022) which achieves comparable performance with our method in age bias mitigation (EI in Table 1), tends to generate multiple individuals in one image to meet the inclusive requirement in the ethical interventions. We argue that these results also demonstrate a misalignment with the given prompt, which specifically requests "**a** {occupation}", implying a single individual.

Table 3: Performance of combining adaptive inclusive tokens to mitigate multi-biases.

| $<i_g>$ | $<i_r>$ | $<i_a>$ | Gender $\mathcal{D}_{\mathcal{KL}}\downarrow$ | Race $\mathcal{D}_{\mathcal{KL}}\downarrow$ | Age $\mathcal{D}_{\mathcal{KL}}\downarrow$ | FID↓ | CLIP↑ |
|---|---|---|---|---|---|---|---|
| | | | 0.3584 | 0.5973 | 0.2319 | 281.12 | **0.2823** |
| ✓ | ✓ | ✓ | **0.1417** | **0.5932** | **0.2272** | **262.95** | 0.2769 |

Table 4: Performance of adaptive inclusive tokens in complex scenes. Reported metrics are on gender bias. Additional prompts are added to "A photo of a $<i_g>$ {occupation}".

| Metrics | $\mathcal{D}_{\mathcal{KL}}\downarrow$ | FID↓ | CLIP↑ | $\mathcal{D}_{\mathcal{KL}}\downarrow$ | FID↓ | CLIP↑ | $\mathcal{D}_{\mathcal{KL}}\downarrow$ | FID↓ | CLIP↑ |
|---|---|---|---|---|---|---|---|---|---|
| Prompts | + "drinking coffee." | | | + "reading a book." | | | + "listening to music." | | |
| SD1.5 | 0.3567 | 266.75 | **0.3116** | 0.3992 | 326.80 | **0.3045** | 0.4246 | **274.67** | **0.3033** |
| Ours | **0.2404** | **264.88** | 0.3082 | **0.2669** | **312.87** | 0.3020 | **0.2195** | 275.71 | 0.3009 |

**Ablation Studies.** Ablation studies are conducted to evaluate the effectiveness of our proposed components. The quantitative results are shown in Tab. 2. As demonstrated, combining two proposed components generally yields the best results except for race bias. We hypothesize that it is because most occupations are biased towards White figures as a homogenous bias distribution, which weakens the effectiveness of our adaptive mapping network.

**Multiple Biases.** To validate the ability of our adaptive inclusive tokens to mitigate multiple stereotypical biases simultaneously, we concatenate previously single-attribute-trained inclusive tokens during inference with prompt: "A photo of a $<i_g>$ $<i_r>$ $<i_a>$ {occupation}". As the results in Tab. 3 demonstrate, the adaptive inclusive tokens can be combined to achieve inclusive generation across various attributes.

**Complex Scene.** We use three additional prompts describing people engaged in various activities to assess the generalizability of our tokens to unseen and more complex prompts. The evaluation is done on the $<i_g>$ token only for a preliminary demonstration. The results are reported in Tab. 4. It appears that our adaptive inclusive token can be effectively generalized to more complex scenarios with comparable text-image alignment and even better image quality.

## 5 CONCLUSION AND LIMITATION

In this work, we study the challenges faced by previous prompt tuning-based bias mitigation approaches. To address the limitation of generalization across different dominant attribute classes, we propose to learn a concept-specific inclusive token through an adaptive mapping network, instead of a fixed one. For the concept drifting issue, we introduce an anchor loss that constrains the impact of the adaptive inclusive token on the final outputs. As a result, our proposed method manages to significantly enhance inclusiveness in T2I generation, with the ability to generalize to unseen concepts and scenarios and mitigate multiple biases simultaneously. In addition, our approach is model-agnostic and can be applied to various T2I models to boost fairness in their outputs.

However, since our target distribution is an even distribution following previous approaches, de-biasing is limited to neutral concepts where all attribute classes are factually correct. For non-neutral concepts, de-biasing may introduce factual errors, such as generating black people as "the pope" with a <race-inclusive> token. Correcting these inaccuracies would require an additional fact-checking process. Nevertheless, these errors may not always be undesirable from an inclusive and creative perspective. The trade-off between accuracy and fairness is a complex issue that requires continuous discussion by researchers and policymakers. We acknowledge this limitation and hope to address it in future work.

We hope our work can bring insights to all who care and work in this field. We believe it should be our effort to make the world a more inclusive place, where every individual regardless of background or identity, can feel valued and respected.

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

APPENDIX

The appendix covers the following contents:

## A  OCCUPATION LISTS

The training occupations are: `["construction worker", "mechanic", "firefighter", "police officer", "engineer", "pilot", "computer programmer", "ceo", "surgeon", "truck driver", "electrician", "scientist", "nurse", "teacher", "secretary", "librarian", "receptionist", "administrative assistant", "childcare worker", "social worker", "dental hygienist", "therapist", "pathologist", "nutritionist"]`

The testing occupations are: `["doctor", "chief", "farmer", "architect", "software developer", "ballet dancer", "yoga instructor", "cosmetologist", "fashion designer", "flight attendant", "artist", "musician", "writer", "photographer", "philosopher", "athlete", "lawyer", "politician", "journalist", "barista", "detective", "security guard", "professor", "sports coach"]`

## B  TRAINING DATA PREPARATION

To construct the training set with known attribute classes, we utilize the Stable Diffusion[4] (SD) model with prompts formatted as "High-quality photo of a/an [attribute] {occupation}". The [attribute] choices include male/female for gender, White/Black/Asian/Middle Eastern/Indian/Latino Hispanic for race, and young/old for age as mentioned in the paper. The {occupation} list is given above. We generate 20 images for each attribute-occupation combination. The RetinaFace detector (Deng et al., 2020) is applied with a confidence filter of 0.97 to ensure a valid face is present in the generated image. Additionally, manual screening of the generated results is conducted to ensure the attribute class and occupation information are correct.

## C  BASELINES

Baseline methods implementations are briefly explained as follows: (1) **Stable Diffusion** (SD1.5) (Rombach et al., 2022): we use it as the original pipeline with base prompt $T(c)$ = "A photo of a {occupation}". (2) **ITI-GEN** (Zhang et al., 2023): we train ITI-GEN on different attributes separately using the additional datasets provided. During generation, it loops over all classes of

---

[4]runwayml/stable-diffusion-v1-5

Table 5: Effect of natural inclusive tokens in gender bias mitigation. The testing prompt is "A photo of a [NIT] {occupation}" and the testing occupations consist of 10 unseen occupations (5 from each gender domination group).

| Natural Inclusive Token | Gender $\mathcal{D}_{\mathcal{KL}} \downarrow$ | FID$\downarrow$ | CLIP$\uparrow$ |
|---|---|---|---|
| None | 0.4783 | 287.55 | 0.2909 |
| "person" | 0.4021 | **282.76** | 0.2936 |
| "individual" | 0.4222 | 287.27 | **0.2960** |
| "diverse" | **0.3122** | 299.14 | 0.2875 |

"A photo of a {NIT} doctor"

Figure 5: Visual effects of natural inclusive token in an occupation-related prompt.

certain attribute. To ensure optimal results as even distribution, we generate an equal number of images for each class, allowing a slightly larger amount of images to be evaluated when the number of classes is not divisible by 100. Since race attribute is not studied in their work, we do race classification on the inclusive skin tone generation and recognize its limitations. (3) **TIME** (Orgad et al., 2023): the model is trained on each test occupation individually since it cannot generalize to unseen concepts. The editing strength parameter $\lambda$ is tuned in choices of $[10, 1, 0.1]$ to select the best one for each occupation. (4) **Fair Diffusion** (Friedrich et al., 2023): we follow the default editing parameters and edit the model to the non-stereotypical class for gender and age attributes with binary classes. For the race attribute consisting of six classes, we loop over all six classes as the editing directions to give a comprehensive result. (5) **Ethical Intervention** (Bansal et al., 2022): we add best-performing intervention "...if all individual can be a {occupation} irrespective of their {attribute}" as reported in their paper to the base prompt. (6) **Fair Mapping** (Li et al., 2023): The model is trained on three attributes separately following the implementation details provided until the training loss converges. (7) Revised **Textual Inversion** (Gal et al., 2022) method is implemented as discussed in Sec. 3.1.

## D    NATURAL INCLUSIVE TOKEN

We have observed the existence of natural inclusive tokens (NIT), which we define as tokens that can be incorporated into an occupation-related prompt to promote a fairer attribute distribution without significantly altering the image structure of the generated results. Through experiments with prompts

formatted as "A photo of a [NIT] {occupation}" on gender bias and 10 unseen occupations[5] from different domination groups following Fig.1(a) of the main paper, we have identified several neutral words that demonstrate the effectiveness of NIT in achieving fairer attribute distributions. The numerical results are shown in Tab. 5. As we can see, NIT can reduce the distribution discrepancy to a certain extent. Albeit not as effective as the adaptive inclusive token that we learned in the main paper, NIT can serve as initialization. The visual effects of NIT are illustrated in Fig. 5. We choose to use "individual" as our initial token to obtain the adaptive inclusive token and experiment with other NITs as initial tokens, as discussed in Sec. E.6.

# E    MORE ABLATION STUDIES

Additional ablation study results are presented. All evaluations are conducted following the evaluation protocols described in the main paper. The face detector is disabled when generating qualitative results to ensure the corresponding images are generated by the same noise latent, facilitating fair and consistent comparisons.

Table 6: Ablation study on the number of adaptive tokens.

| No. of Inclusive Tokens | Gender $\mathcal{D}_{\mathcal{KL}} \downarrow$ | Race $\mathcal{D}_{\mathcal{KL}} \downarrow$ | Age $\mathcal{D}_{\mathcal{KL}} \downarrow$ |
|:---:|:---:|:---:|:---:|
| 1 (Ours) | **0.1298** | **0.3625** | 0.2168 |
| 2 | 0.1560 | 0.3893 | 0.2337 |
| 3 | 0.1575 | 0.3646 | **0.2131** |

## E.1    NUMBER OF INCLUSIVE TOKENS

We try to learn different numbers of tokens to carry our inclusive requirement. Tab. 6 demonstrates the effectiveness of a single adaptive token in achieving a more inclusive distribution in the output.

## E.2    ADAPTIVE MAPPING NETWORK COMPLEXITY

We employ the transformer architecture for the adaptive mapping network and explore variations in transformer complexities, as shown in Tab. 7. The results suggest that our adaptive inclusive token effectively reduce stereotypical bias in text-to-image (T2I) results regardless of the complexity of the adaptive mapping network. Generally, employing more transformer blocks yields better inclusiveness in gender bias but shows less effectiveness in mitigating race bias. This observation is consistent with Tab.2 of the main paper that the adaptive mapping module is less effective in race de-bias since most occupations are biased toward White individuals, resulting in a single primary domination group. On the other hand, age bias mitigation performance is insensitive to different network complexities.

Table 7: Ablation study on the complexity of the adaptive mapping network.

| Transformer Blocks | Attention Heads | Gender $\mathcal{D}_{\mathcal{KL}} \downarrow$ | Race $\mathcal{D}_{\mathcal{KL}} \downarrow$ | Age $\mathcal{D}_{\mathcal{KL}} \downarrow$ |
|:---:|:---:|:---:|:---:|:---:|
| SD1.5 Original Pipeline | | 0.3584 | 0.5973 | 0.2319 |
| 4 | 6 | **0.1298** | 0.3625 | 0.2168 |
| 2 | 6 | 0.1635 | **0.3156** | 0.2026 |
| 4 | 8 | 0.1310 | 0.3550 | **0.1989** |
| 2 | 8 | 0.1820 | 0.3220 | 0.2127 |

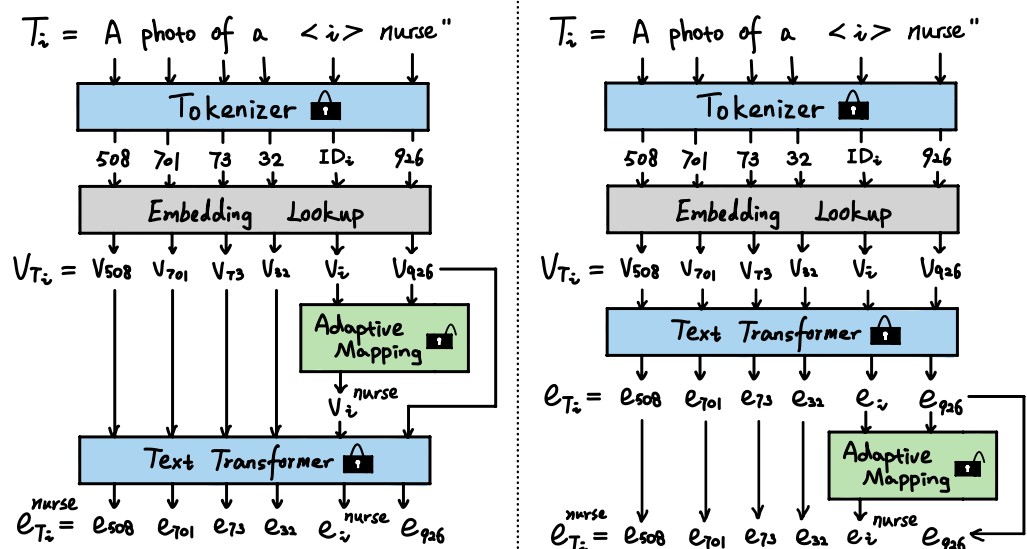

Figure 6: Various placements of adaptive mapping module. **Left**: before text transformer (inclusive embedding in token embeddings). **Right**: after text transformer (inclusive embedding in text embeddings).

Table 8: Ablation study on the placement of the adaptive mapping module.

| Placement of Adaptive Mapping | Gender $\mathcal{D}_{\mathcal{KL}} \downarrow$ | Race $\mathcal{D}_{\mathcal{KL}} \downarrow$ | Age $\mathcal{D}_{\mathcal{KL}} \downarrow$ |
|---|---|---|---|
| Before Text Transformer | **0.1298** | **0.3625** | **0.2168** |
| After Text Transformer | 0.3519 | 0.5644 | 0.2667 |

### E.3 PLACEMENT OF ADAPTIVE MAPPING MODULE

The placement of the adaptive inclusive embedding within the text model can significantly impact its effectiveness in bias mitigation. To explore the effects of placing the inclusive embedding in different locations, we conduct experiments by placing the adaptive mapping module before and after the text transformer, as illustrated in Fig. 6. The results are shown in Tab. 8. Our findings indicate that adding the adaptive mapping module after the text transformer reduces the influence of the inclusive embedding on the final outputs, resulting in reduced effectiveness in bias mitigation.

We further investigate the impact of embeddings before and after text transformer with two sets of simple experiments. Here, we denote the textual embeddings before the text transformer as token embeddings $v_{T_i}$, and the textual embeddings after the text transformer as text embeddings $e_{T_i}$. Firstly, for a neutral occupation-related prompt "A photo of a {occupation}" where no gender class is specified, we find that most of the output images still follow the occupation's stereotypical gender even if the embedding of the occupation's non-stereotypical gender is added after the text transformer. This observation suggests that the gender bias associated with the occupation is spread to the whole text embeddings via text transformer, and late indication in text embedding does not play a decisive role in the outcome. Secondly, to further validate this finding, we modify the prompt to include a non-stereotypical gender class, *e.g.*, "A photo of a male nurse", and subsequently replace the text embedding of "male" with that of "female" (obtained from another prompt, "A photo of a female nurse") after the text transformer. Surprisingly, the resulting nurse images continue to predominantly exhibit the male gender. This result indicates that the attribute indicators have a greater impact on the final outputs when incorporated into the token embeddings, as the effect of the indicators can propagate throughout the entire text embedding via the transformer architecture. This

---

[5]["doctor", "chief", "farmer", "architect", "software developer", "ballet dancer", "yoga instructor", "cosmetologist", "fashion designer", "flight attendant"]

Table 9: Ablation study on the number of training samples.

| Samples per Attribute-Occupation | Gender $\mathcal{D}_{\mathcal{KL}} \downarrow$ | Race $\mathcal{D}_{\mathcal{KL}} \downarrow$ | Age $\mathcal{D}_{\mathcal{KL}} \downarrow$ |
|---|---|---|---|
| 20 | **0.1298** | 0.3625 | **0.2168** |
| 10 | 0.2641 | **0.3251** | 0.2177 |
| 5 | 0.1542 | 0.3636 | 0.2907 |

Table 10: Ablation study on image sources.

| Image Source | Gender $\mathcal{D}_{\mathcal{KL}} \downarrow$ | FID↓ | CLIP↑ |
|---|---|---|---|
| SD1.5 (888) | **0.1593** | **263.67** | **0.2795** |
| Online (888) | 0.2898 | 269.65 | 0.2768 |

finding aligns with our observation that the adaptive inclusive token affects the attributes of the final results more effectively when the adaptive mapping module is placed before the text transformer.

### E.4 NUMBER OF TRAINING SAMPLES

The effect of the number of training samples is shown in Tab. 9. It can be seen that though the bias mitigation performance does not improve linearly with more training samples per attribute-occupation combination, 20 samples per combination generally yields the best results.

### E.5 IMAGE SOURCES

We investigate the impact of different image sources on the bias mitigation performance. Due to the difficulty in finding real images for certain combinations of attribute classes and occupations, this study is limited to gender bias. We use `clip-retrieval` (Beaumont, 2022) to retrieve 20 images per attribute-occupation combination from the LAION-400M dataset (Schuhmann et al., 2021) and manually screen the retrieved images to retain those with the correct attribute classes and visual concepts associated with the occupation. To ensure balanced training data, we maintain a consistent number of images for each gender within each occupation category, resulting in 888 real images in the training set. To make a fair comparison, we align the number of synthetic data generated by SD1.5 with real data in each attribute-occupation combination, resulting in 888 synthetic images as well. The results are shown in Tab. 10. In conclusion, the T2I results have better inclusiveness, image quality, and textual alignment when the adaptive inclusive token is trained with synthetic images generated by the same SD model.

### E.6 INITIAL TOKEN

As discussed in Sec. D, we explore various NITs as the initial token to assess their impact on bias mitigation performance. From Tab. 11, we can see that all initial tokens demonstrate effective bias mitigation across all three bias attributes. Furthermore, if we deviate from the conventional practice of initializing the pseudo token to an existing word (Gal et al., 2022) by employing randomly initialized weights, our method's performance remains comparable. This demonstrates the robustness and effectiveness of our adaptive mapping network to predict a reasonable inclusive token regardless of the token initialization.

### E.7 RESULTS ON SD2.1 AND SDXL

The generalization ability of our method on more advanced SD models SD2.1[6] (Rombach et al., 2022) and SDXL[7] (Podell et al., 2024) is evaluated. Tab. 12 showcases the generalizability of our method to SD2.1 model. Following the data preparation procedure outlined in Sec. B, training data is generated using the SDXL model. The adaptive mapping module is integrated into the first text

---

[6]stabilityai/stable-diffusion-2-1

[7]stabilityai/stable-diffusion-xl-base-1.0

Table 11: Ablation study on different NIT as the initial token.

| Initial Token | Gender $\mathcal{D}_{\mathcal{KL}}\downarrow$ | Race $\mathcal{D}_{\mathcal{KL}}\downarrow$ | Age $\mathcal{D}_{\mathcal{KL}}\downarrow$ |
|---|---|---|---|
| "individual" | 0.1298 | 0.3625 | 0.2168 |
| "person" | 0.1410 | **0.3518** | 0.2273 |
| "diverse" | 0.1413 | 0.3699 | 0.2070 |
| Random Weights | **0.1288** | 0.3839 | **0.1865** |

model of SDXL, while the second text model remains unchanged. The results are presented in Tab. 13. The qualitative results are shown in Fig. 7, 8, 9. Both quantitative and qualitative results demonstrate that our adaptive inclusive token method can be generalized to a more advanced SD model to achieve more inclusive T2I outcomes.

Table 12: Performance on Stable Diffusion 2.1.

| Diffusion Model | Gender $\mathcal{D}_{\mathcal{KL}}\downarrow$ | Race $\mathcal{D}_{\mathcal{KL}}\downarrow$ | Age $\mathcal{D}_{\mathcal{KL}}\downarrow$ |
|---|---|---|---|
| SD2.1 (Rombach et al., 2022) | 0.4166 | 0.5819 | 0.2062 |
| SD2.1 with ours | **0.3074** | **0.5722** | **0.1934** |

Table 13: Performance on SDXL.

| Diffusion Model | Gender $\mathcal{D}_{\mathcal{KL}}\downarrow$ | Race $\mathcal{D}_{\mathcal{KL}}\downarrow$ | Age $\mathcal{D}_{\mathcal{KL}}\downarrow$ |
|---|---|---|---|
| SDXL(Podell et al., 2024) | 0.4919 | 0.6796 | 0.1965 |
| SDXL with ours | **0.1816** | **0.3986** | **0.1924** |

## F    MORE QUALITATIVE RESULTS

More qualitative results of applying our adaptive inclusive token on the SD1.5 model are presented in Fig. 10, 11, 12.

"A photo of a doctor"

SDXL

SDXL
With Ours

"A photo of a yoga instructor"

SDXL

SDXL
With Ours

Figure 7: Qualitative results of gender bias mitigation on SDXL. All images are generated with the same random seed.

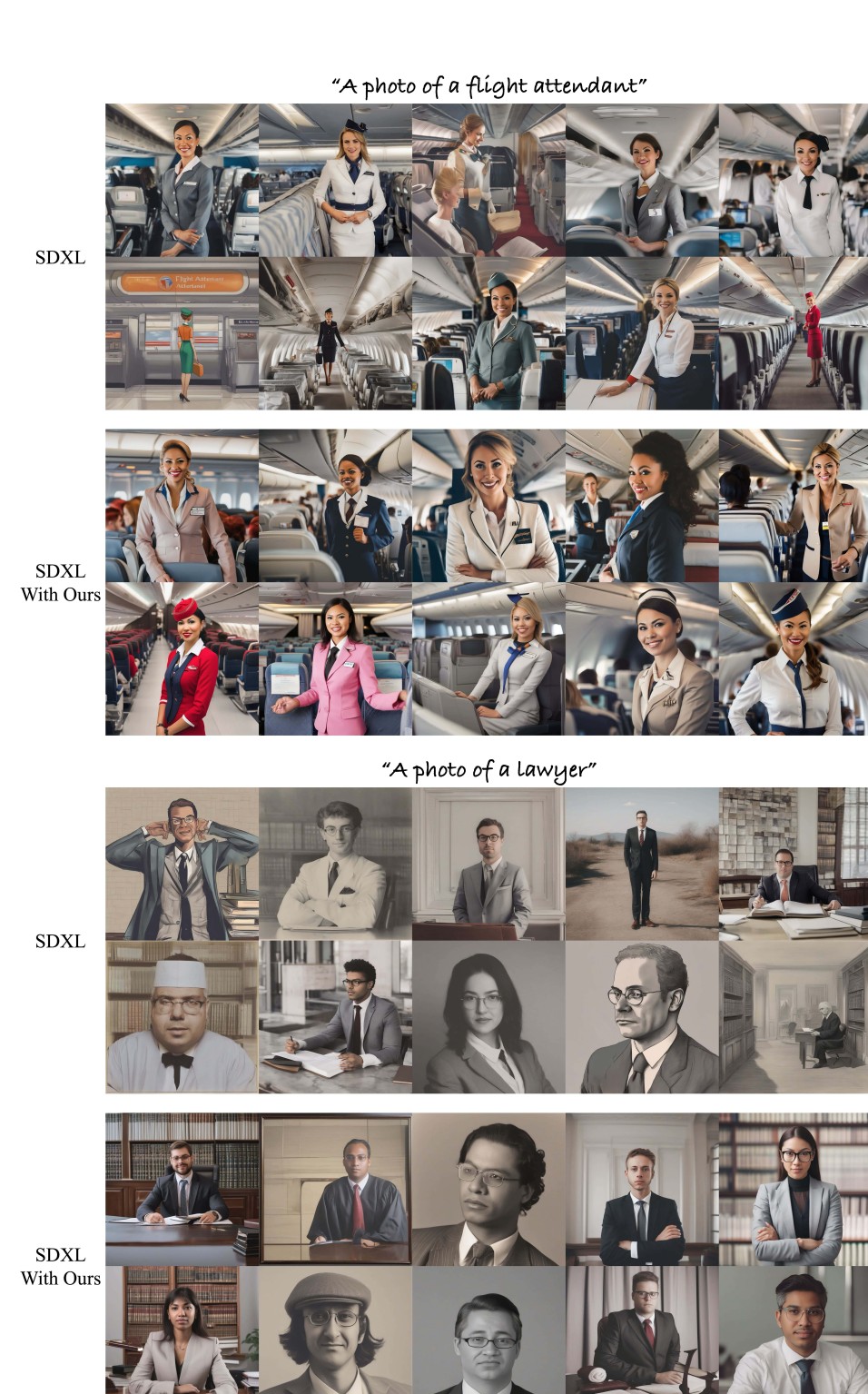

Figure 8: Qualitative results of race bias mitigation on SDXL. All images are generated with the same random seed.

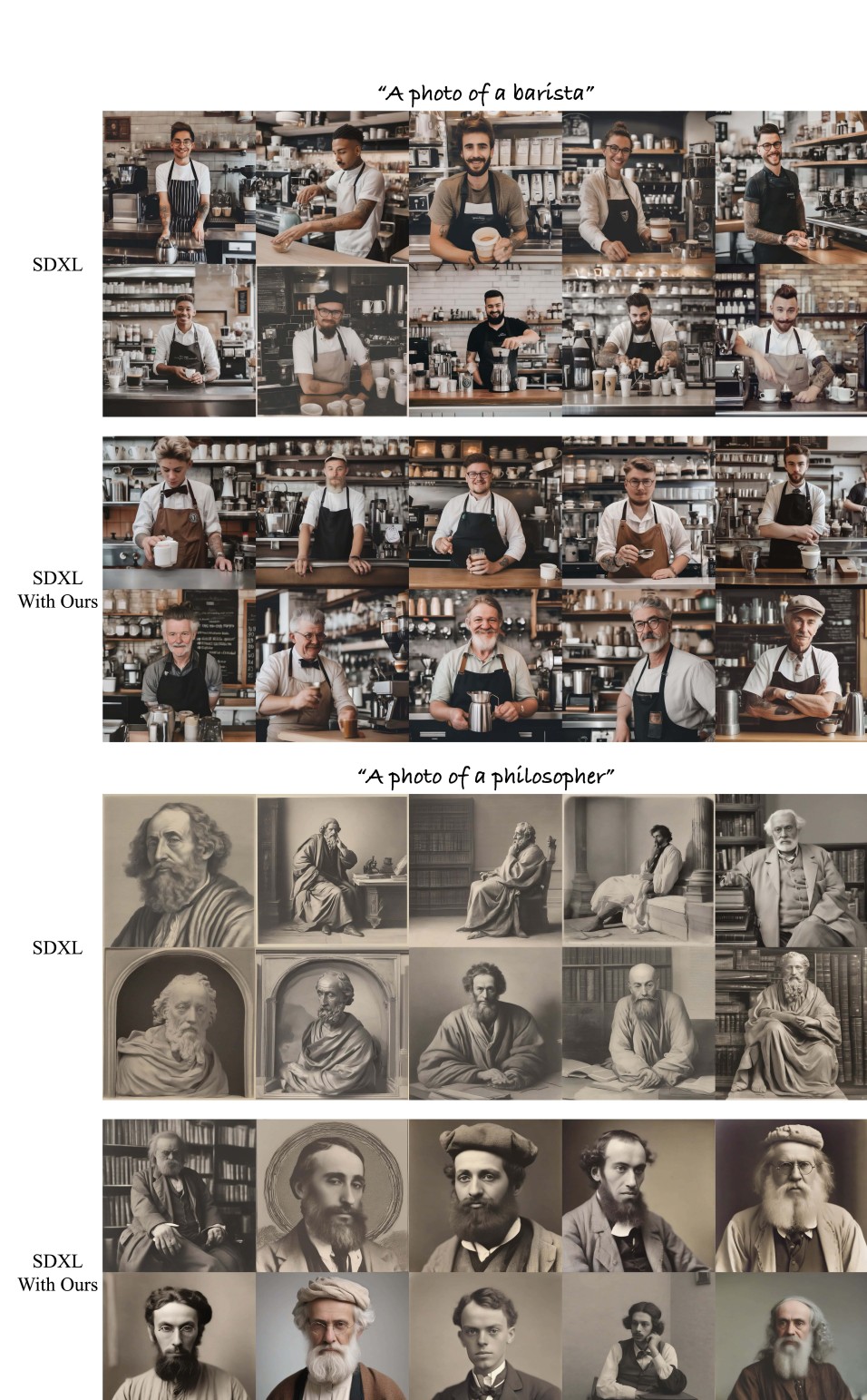

Figure 9: Qualitative results of age bias mitigation on SDXL. All images are generated with the same random seed.

"A photo of a software developer"

SD1.5

SD1.5
With Ours

"A photo of a cosmetologist"

SD1.5

SD1.5
With Ours

Figure 10: Qualitative results of gender bias mitigation on SD1.5. All images are generated with the same random seed.

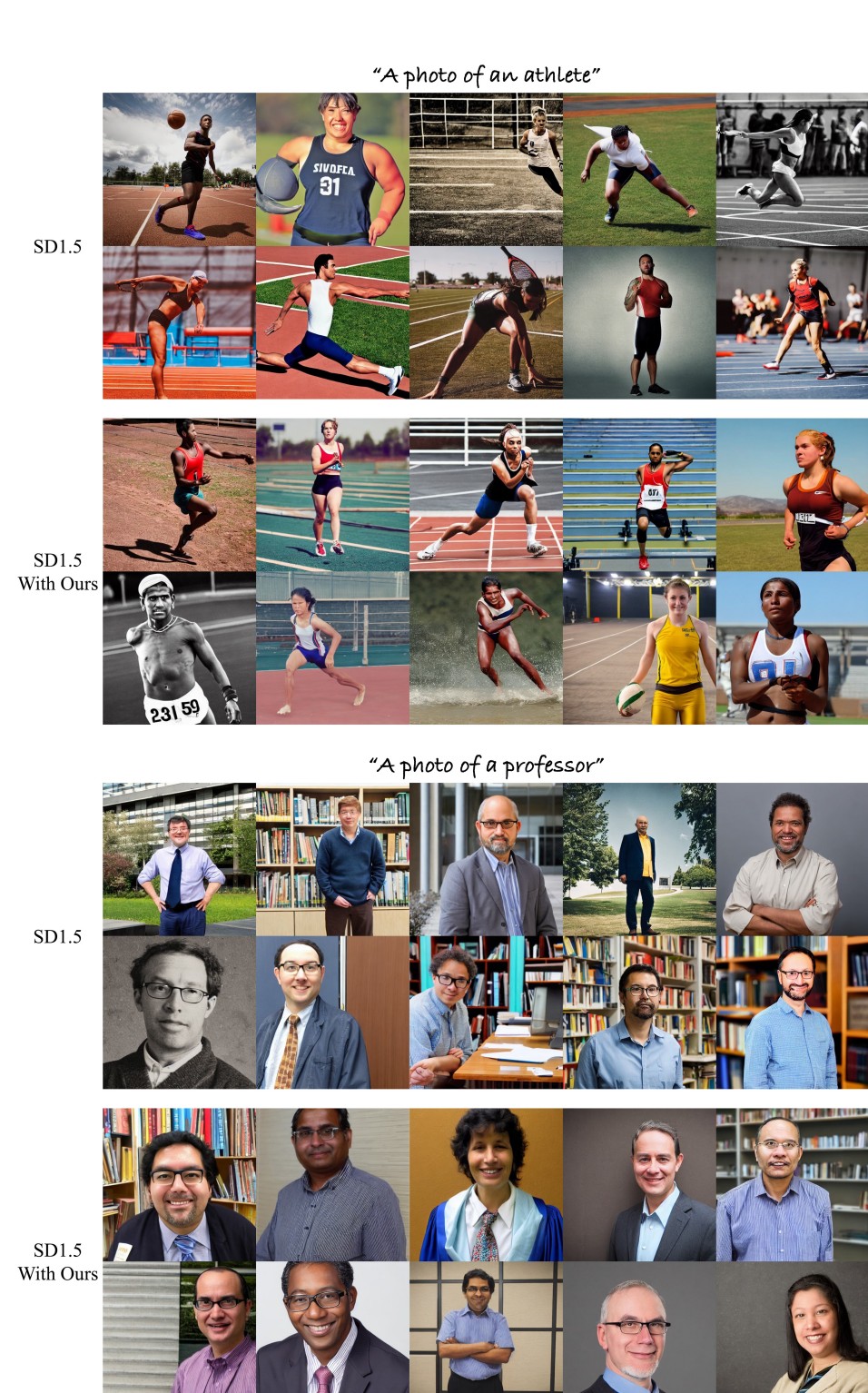

Figure 11: Qualitative results of race bias mitigation on SD1.5. All images are generated with the same random seed.

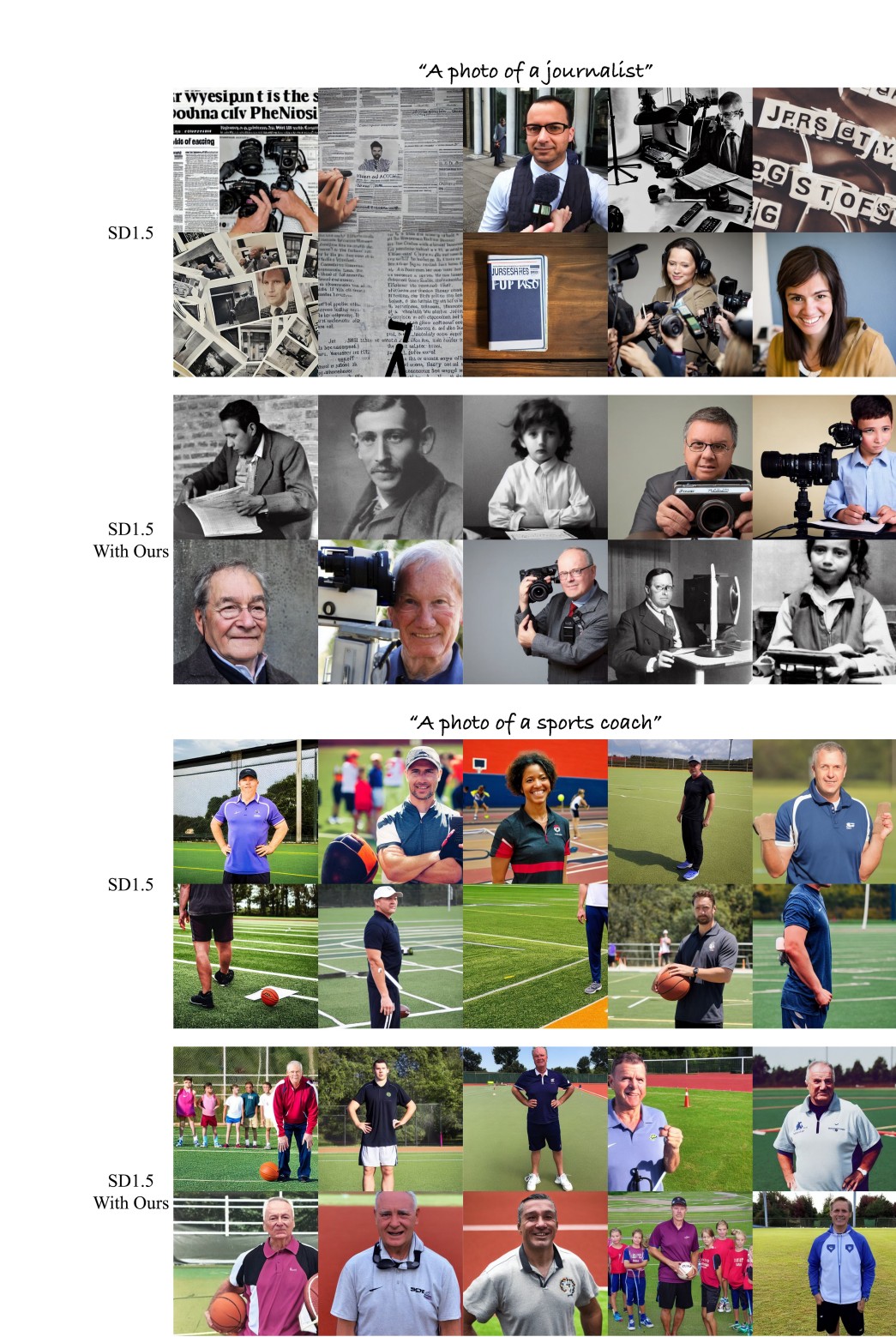

Figure 12: Qualitative results of age bias mitigation on SD1.5. All images are generated with the same random seed.

