# OpenReview forum: "AITTI: Learning Adaptive Inclusive Token for Text-to-Image Generation"
_ICLR.cc/2025/Conference — Submitted to ICLR 2025_

### Official Review · Reviewer_kTpz · 2024-10-31

**Soundness:** 2
**Presentation:** 2
**Contribution:** 2
**Rating:** 5
**Confidence:** 4

**Summary:**

This paper proposes a bias mitigation approach for T2I generation using adaptive inclusive tokens, enhancing fairness without needing attribute specifications. By employing an adaptive mapping network and anchor loss, the approach effectively mitigates biases while maintaining quality and text alignment. The method is versatile but has limitations when handling non-neutral concepts and potential semantic drift.

**Strengths:**

1. The paper is well-written and reads clearly.
2. Learning an adaptive mapping network to generate inclusive tokens specific to different occupation is reasonable and justified.
3. Experimental results effectively demonstrate the method’s effectiveness.

**Weaknesses:**

1. Calculating anchor loss requires predefined information on non-dominant bias attributes for each occupation in the training data, which necessitates human pre-determination.
2. The paper primarily focuses on biases introduced by the text encoder and does not examine potential biases inherent to the diffusion model itself. However, text embedding may be attacked[1]. The proposed method may become ineffective after being attacked.
3. The reported FID metric is in the range of 200–300, whereas this metric usually falls within the magnitude of 10^1. This discrepancy may result from insufficient statistical sampling, casting doubt on whether FID accurately reflects image quality here. ImageReward [2] or PickScore [3] might be fairer metrics in this context.

[1] Reliable and Efficient Concept Erasure of Text-to-Image Diffusion Models

[2] ImageReward: Learning and Evaluating Human Preferences for Text-to-Image Generation

[3] Pick-a-Pic: An Open Dataset of User Preferences for Text-to-Image Generation

**Questions:**

Please see Weaknesses.

---

> ### Author Response · Authors · 2024-11-22
>
> We thank Reviewer kTpz for acknowledging the clarity of our writing, the justification of our adaptive mapping network, and the experimental results. The concerns raised by the reviewer are address below:
>
> \
> **Calculating anchor loss requires predefined information on non-dominant bias attributes for each occupation**
>
> We would like to clarify that our anchor loss does not require pre-defined information about non-dominant bias attributes. The anchor prompt is derived from the ground-truth attribute class of the training image, which is readily accessible. Our method operates without requiring prior knowledge of the original bias distribution of the target concepts, ensuring its applicability without human pre-determination.
>
> \
> **Inherent bias within the diffusion model**
>
> We appreciate the reviewer’s observation. As with most previous bias mitigation approaches, our work primarily focuses on addressing biases introduced to the diffusion model by text embeddings, as these are more identifiable and actionable. The inherent biases within the diffusion model itself are challenging to isolate and remain an open research question.
>
> Regarding potential attacks such as concept erasing, we acknowledge that this could impact the effectiveness of our method. However, robustness against adversarial attacks falls outside the scope of our current research, which aims to mitigate bias in standard, unaltered setups. This is an important direction for future exploration.
>
> \
> **About FID metric**
>
> We thank the reviewer for suggesting alternative metrics for evaluating image quality. Following the suggestion, we evaluated the image quality of SD1.5, Fair Mapping (FM), and our method using the ImageReward [1] metric. We computed the average reward score across 2,400 images from SD1.5 (24 test occupations, each with 100 images) and 7,200 images for both FM and our method (24 test occupations, each with 100 images for gender, race, and age bias mitigation). The results are as follows:
>
> | Metric          | SD1.5  | FM      | Ours   |
> |------------------|--------|---------|--------|
> | ImageReward (↑) | 0.5345 | -0.9078 | 0.3833 |
>
> ImageReward comprehensively considers human preference, image fidelity, and text-image alignment when assessing image quality. These results demonstrate that our method achieves fairer results while maintaining acceptable image quality, especially when compared to FM, which exhibits significant degradation in image quality.
>
> [1] ImageReward: Learning and Evaluating Human Preferences for Text-to-Image Generation. NeurIPS, 2023.

---

> > ### Comment · Reviewer_kTpz · 2024-11-28
> >
> > Thanks for the author's response, some of my concerns have been addresed. However, I still have concerns about the robustness of the proposed method:
> > 1. According to the author's clarification, social biases in generated images mainly come from the inherent biases in SD1.5's CLIP text encoder. Would using a stronger text encoder eliminate this bias? For example, DeepFloy-IF and SD3 use T5 as their text encoders, and recent work has started to use LLMs as text encoders. Additionally, I agree with other reviewers' concerns about the motivation of the paper;
> > 2. Based on ImageReward metrics, it seems that the proposed method compromises image aesthetic quality and alignment with prompts (compared to SD1.5), which may be more pronounced in complex prompts.

---

> > > ### Author Response · Authors · 2024-11-28
> > >
> > > We thank Reviewer kTpz for their thoughtful follow-up and for highlighting these important concerns. We appreciate the opportunity to provide additional clarifications.
> > >
> > > \
> > > First, regarding the use of stronger text encoders, we would like to clarify that stronger encoders are not necessarily better in terms of fairness. These models are typically trained on larger datasets, which may introduce or amplify biases present in the data. Furthermore, we did not claim that "social biases in generated images mainly come from the inherent biases in SD1.5's CLIP text encoder." Inherent biases exist within both the diffusion model and the text encoder. However, biases in the diffusion model can be harder to distinguish in unconditional generation and tend to have less societal impact. In contrast, text-to-image generation using language clearly reveals social biases due to the explicit connections between prompts and outputs, making these biases more apparent and potentially more harmful given the widespread use of language-driven AI systems. This is why our method focuses on mitigating biases stemming from text-based inputs.
> > >
> > > \
> > > Second, regarding concerns about motivation, we reiterate that our work addresses biases in neutral prompt scenarios, which is a critical aspect of fairness in generative models. As discussed in prior responses, fairness cannot be fully addressed by explicit prompt conditioning, as this merely evaluates prompt-following accuracy rather than the model's unbiased behavior. Importantly, as noted in ITI-GEN, using explicit conditions in prompts may not always yield optimal results due to linguistic ambiguity or model misrepresentation. These limitations underscore the need for approaches like ours, which reduce reliance on explicit prompt specification and ensure more inclusive outputs under neutral conditions. Our method thus addresses a significant gap in current approaches and provides practical solutions for fairness.
> > >
> > > \
> > > Finally, on the issue of ImageReward metrics, we acknowledge that our method results in a slight compromise in aesthetic quality compared to SD1.5, as indicated by ImageReward scores. However, this is a minor trade-off compared to the large degradation observed in Fair Mapping, as shown in our results. Additionally, we direct the reviewer to our qualitative results in Figure 3, 4, 7, 8, 9, 10, 11, 12, which clearly demonstrate that the image quality of our method does not degrade compared to the original SD model. These qualitative examples highlight that our method effectively balances fairness with high-quality image generation.
> > >
> > > \
> > > We hope these clarifications address your remaining concerns and highlight the robustness and necessity of our method. We respectfully ask you to consider these points in your final assessment, and we sincerely appreciate your valuable feedback and engagement with our work.

---

### Official Review · Reviewer_JdSh · 2024-11-02

**Soundness:** 3
**Presentation:** 2
**Contribution:** 2
**Rating:** 5
**Confidence:** 4

**Summary:**

The paper presents a novel approach to mitigate stereotypical biases in text-to-image (T2I) generation models. The authors introduce a method for learning adaptive inclusive tokens that can adjust the attribute distribution of the generated images to be more equitable, without the need for explicit attribute specification or prior knowledge of the bias distribution. The core of their approach is a lightweight adaptive mapping network that customizes inclusive tokens for concepts that are targeted for de-biasing, which enhances the generalizability of the tokens to new, unseen concepts. The method is evaluated on the Stable Diffusion framework and demonstrates improved performance over existing bias mitigation techniques, showing its effectiveness in reducing stereotypical biases in T2I generation.

**Strengths:**

1. The paper introduces a novel method for reducing biases in text-to-image generation, which is critical to AI ethics and fairness. The concept of learning adaptive inclusive tokens that can shift attribute distributions in generative outputs is a creative solution.
2. The method's ability to generalize to unseen concepts and handle multiple attributes is a major advantage, showcasing its potential for broad application across various domains.

**Weaknesses:**

1. While the authors claim to have developed a lightweight adaptive mapping network to address bias mitigation, the training time of nearly one hour suggests that it may not be as lightweight as initially proposed. This process raises concerns about the practicality and efficiency of the solution, especially in contexts where rapid deployment or real-time adjustments are required.
2. While the authors demonstrate quantitative improvements over the baseline in comparative analysis, the qualitative examples presented in the supplementary material show that the method is not entirely effective, with many prompts still generating images with a predominantly single gender. This inconsistency between quantitative and qualitative results casts doubt on the overall effectiveness of the method and its ability to achieve the desired inclusiveness in text-to-image generation.
3. Overall, I think the authors' approach of training to shift certain biased concepts towards less ambiguous representations using a better mapping network, akin to a new token, is quite straightforward. However, as mentioned earlier, I am concerned about whether such a simple method can be truly effective.

**Questions:**

1. The main metric $D_{kl}$ for attribute distribution demonstrates that the original SD1.5 outperforms SD2.1 and SDXL (SD1.5>SD2.1>SDXL). This is not consistent with the qualitative results. Does it mean the metric there is a problem with the metric itself?
 2. Why does a single token perform better than multiple tokens (with higher complexity)?

---

> ### Author Response · Authors · 2024-11-22
>
> We thank Reviewer JdSh for recognizing the novelty and importance of our method. We also appreciate your acknowledgment of our creative use of adaptive inclusive tokens and the method's ability to generalize to unseen concepts and handle multiple attributes. The concerns are addressed below:
>
> \
> **Practicality and efficiency**
>
> We thank the reviewer for their concern about the practicality and efficiency of our method. Compared to previous approaches that require extensive fine-tuning of diffusion models (e.g., ~48 hours on 8 NVIDIA A100 GPUs for a single attribute [1]) or large-scale external network training [2], our method is significantly more efficient. It requires only one hour on a single GPU using a small set of easily obtainable SD-generated images. This demonstrates that our adaptive mapping network and inclusive token training are lightweight in the context of bias mitigation.
>
> Furthermore, once trained, the adaptive inclusive token generalizes to unseen concepts across different domination groups, eliminating the need for retraining. This enables reusability and makes our method practical and efficient for real-world applications.
>
> [1] Fine-tuning Text-to-Image Diffusion Models for Fairness. ICLR, 2024. \
> [2] ITI-GEN: Inclusive Text-to-Image Generation. ICCV, 2023.
>
> \
> **Qualitative results**
>
> We thank the reviewer for recognizing our quantitative improvements over the baseline. However, we are unclear about the specific qualitative examples being referred to as “many prompts still generating images with a predominantly single gender.” In all qualitative examples addressing gender bias mitigation (Figures 3, 7, and 10), our method consistently demonstrates clear improvements in gender diversity compared to the baseline.
>
> For figures targeting other biases, such as race or age, it is reasonable that gender bias may persist, as those experiments were not designed to address gender bias specifically. We did not claim that our method mitigates all types of bias using single-attribute inclusive token; rather, we focus on the target attribute for each inclusive token. We hope this clarifies the observed discrepancy.
>
> \
> **Question about effectiveness**
>
> We thank the reviewer for recognizing the simplicity of our approach. As demonstrated by both quantitative and qualitative results, our method effectively mitigates the attribute bias of interest. Furthermore, Table 3 in the main paper highlights the flexibility of our method, showcasing how stacking different inclusive tokens can address multiple biases simultaneously. This demonstrates that our approach, while straightforward, is both effective and versatile.
>
> \
> **Clarification on $D_{KL}$ metric**
>
> We appreciate the reviewer’s observation regarding the $D_{KL}$ values of the original SD1.5, SD2.1, and SDXL models. It is indeed noteworthy that SD1.5 achieves the most inclusive results, with $D_{KL}$ rankings for gender bias as follows: 0.3584 (SD1.5), 0.4166 (SD2.1), and 0.4919 (SDXL). However, we are unclear about which specific qualitative results the reviewer finds inconsistent with the $D_{KL}$ metric.
>
> It is important to note that $D_{KL}$ solely measures the fairness of the model outputs and does not account for image quality. A possible explanation for the observed bias in more advanced models could be the larger and potentially more biased datasets used for training. Additionally, we would like to highlight that with our adaptive inclusive token, all three models demonstrate significant improvements in $D_{KL}$, further validating the effectiveness of our method.
>
> \
> **A single token vs. multiple tokens**
>
> We had similar initial intuitions that multiple tokens, with higher complexity, might outperform a single token due to their potentially greater expressiveness. However, upon analyzing the results, we observed that multiple tokens often learned redundant concepts from the reference set, as our method does not explicitly instruct them on what to capture. Additionally, the reference training sets we use are relatively small, making multiple tokens prone to overfitting due to their higher expressiveness. In this case, a single token proves sufficient and even outperforms multiple tokens by avoiding both redundancy and overfitting.

---

> > ### Comment · Reviewer_JdSh · 2024-11-24
> > **Feedback to the authors' comments**
> >
> > Thank you for your feedback to my concerns. My concerns about efficiency are addressed. On the other hand, I agree with Reviewer QTbU on the concerns about the evaluation accuracy. On the hand, I am not sure if the bias in the unconditional generation can be easily resolved by giving clear conditions. As the yoga instructor example in Figure 7, when specifying the gender, SDXL can easily give the correct output without gender bias. It makes me waver on the motivation of this work (I recommend the authors validate their method on attribute compositionality). In summary I would like to keep my ratings.

---

> ### Author Response · Authors · 2024-11-24
>
> We sincerely thank Reviewer JdSh for their thoughtful follow-up and for acknowledging that we addressed concerns regarding efficiency.
>
> \
> Regarding the **evaluation accuracy** concerns raised by Reviewer QTbU, we would like to emphasize that the CLIP zero-shot classifier is widely used in the bias mitigation literature [1,2,3,4,5] for classifying sensitive attributes thanks to its robust performance and broad applicability. While it is true that CLIP is trained on real-world datasets and may reflect certain biases, our experiments demonstrate that it provides sufficiently accurate attribute classifications for evaluating fairness, as shown in our responses to Reviewer QTbU. Furthermore, its widespread adoption ensures comparability with prior work in this area. While we acknowledge that no single metric is perfect, we believe our experimental results, including both quantitative and qualitative evaluations, provide a comprehensive view of the effectiveness of our method.
> \
> [1] ITI-GEN: Inclusive Text-to-Image Generation. ICCV, 2023. \
> [2] How well can Text-to-Image Generative Models understand Ethical Natural Language Interventions? EMNLP, 2022 \
> [3] Fair Text-to-Image Diffusion via Fair Mapping. arXiv, 2023. \
> [4] Balancing Act: Distribution-Guided Debiasing in Diffusion Models. CVPR, 2024. \
> [5] FairRAG: Fair Human Generation via Fair Retrieval Augmentation. CVPR, 2024.
>
> \
> On the question of **bias in unconditional generation versus specifying clear conditions**, we thank the reviewer for raising this important concern, as it allows us to further clarify and validate the motivation for our work. The key motivation of our method lies in addressing inherent biases in text-to-image generation when prompts are neutral and no explicit attribute is specified. This is a critical issue for fairness because many real-world use cases involve users providing high-level prompts without detailed specifications. Relying on users to explicitly define attributes (e.g., specifying "female doctor" or "male nurse") shifts the responsibility of fairness onto the user and does not address the underlying bias in the model. The definition of inclusiveness for a generative model focuses on how inclusive the outputs are when given no explicit attribute conditions. Generating female doctors when explicitly instructed with "female doctor" does not reflect the fairness of the model but rather its prompt-following accuracy. Inclusiveness is better measured by the model's ability to generate balanced results when the prompt is neutral.
>
> \
> Regarding the **yoga instructor example** in Figure 7, we would like to clarify that there is no gender specified in the prompt. The upper SDXL results are generated with the prompt "A photo of a yoga instructor," while the bottom SDXL With Ours results are generated with the prompt "A photo of a <gender-inclusive> yoga instructor." These results clearly demonstrate the effectiveness of our trained adaptive inclusive token, <gender-inclusive>, in bringing more male figures into the generation of yoga instructors, addressing the observed bias in the original model.
>
> \
> Furthermore, to address the reviewer's suggestion regarding **attribute compositionality**, we have already validated our method in Table 3. By leveraging different inclusive tokens trained for distinct sensitive attributes, our method successfully mitigates multiple biases simultaneously, demonstrating the flexibility and robustness of our approach. This capability further underscores the motivation of our work to create more inclusive generative models capable of addressing a variety of biases in diverse scenarios.
>
> \
> In summary, our work seeks to ensure that generative models produce fair and balanced outputs in scenarios where neutrality is expected. This fills an important gap in existing methods and remains distinct from explicitly conditioned generation tasks, where prompt-following accuracy is evaluated instead of fairness. We sincerely hope that our detailed clarifications and additional validations address the reviewer’s concerns and highlight the strengths of our method in tackling bias in generative models. By emphasizing the importance of inclusiveness in neutral prompts and demonstrating the effectiveness of our approach through attribute compositionality, we aim to showcase the significance and robustness of our work. We kindly ask the reviewer to consider these points when reassessing their score.

---

### Official Review · Reviewer_SjSS · 2024-11-04

**Soundness:** 3
**Presentation:** 2
**Contribution:** 2
**Rating:** 5
**Confidence:** 3

**Summary:**

This pape addresses the critical issue of stereotype bias in text-to-image generation models. The authors propose an innovative method that adjusts the attribute distribution of the final generated output by learning adaptive inclusive tokens. This approach does not require explicit specification of attributes or prior knowledge of bias distribution. Instead, it customizes concept-specific inclusive tokens through a lightweight adaptive mapping network. The method introduces an anchoring loss to constrain the influence of adaptive inclusive tokens on the final output, thereby enhancing the fairness and inclusivity of the generated results without compromising the consistency between the text and the generated images.

**Strengths:**

The paper introduces a novel approach by using adaptive inclusive tokens to mitigate bias in text-to-image generation models without the need for explicit attribute specification or prior knowledge of bias distribution, sounds interesting.

Experimental results demonstrate that the proposed method outperforms previous bias mitigation techniques in scenarios without attribute specification.

The method seems generalizes well and this paper is easy to read.

**Weaknesses:**

Potential Introduction of Factual Errors: The proposed method may introduce factual inaccuracies when dealing with non-neutral concepts. For example, generating an image of a female U.S. president, which does not align with historical facts. The authors do not discuss or evaluate the conflict between accuracy and fairness in their approach.

Lack of Motivation and Theoretical Foundation: The adaptive mapping network proposed by the authors lacks a clear motivation and theoretical discussion. This makes it difficult to understand the underlying principles and rationale behind the approach.

Implementation Details: While the paper outlines the overall framework of the method, the description of the implementation is vague, particularly regarding the specific structure and training process of the adaptive mapping network. The diagrams provided are unclear and fail to effectively convey the authors' intentions.

Quality of Generated Models: There is a lack of effective discussion and comparison regarding the quality of the generated models. The impact of the prompts, models, and data volume on the final model is not adequately addressed.

Limited Experimental Setup: The experimental setup is relatively limited, especially in terms of evaluating performance in complex scenarios and across different datasets. This restricts the understanding of the method's robustness and applicability.

Practical Application Value: The paper lacks a discussion on the practical application value of the proposed method. It is unclear how the generated images can be ensured to be both fair and factually accurate.

**Questions:**

Provide a deeper exploration of the theoretical foundation and motivation behind the adaptive mapping network and anchor loss. Discuss how these components compare to existing bias mitigation techniques.

Include a more detailed description of the structure and training process of the adaptive mapping network in the paper. This will help other researchers better understand and replicate the method.

Enhance the clarity and readability of the figures and diagrams in the paper to effectively convey the authors' intentions.

Increase the number of experiments in complex scenarios. Discuss the impact of fairness on accuracy and how to mitigate such issues.

Provide some practical application cases to demonstrate the effectiveness and potential challenges of the method in real-world scenarios. This will help readers better understand the practical application value of the method.

---

> ### Author Response · Authors · 2024-11-22
>
> We thank Reviewer SjSS for reviewing our paper. The concerns raised are addressed below:
>
> \
> **Introduction of factual error**
>
> We thank the reviewer for raising this concern. As discussed in Section 5 of the main paper, we acknowledge the potential introduction of factual inaccuracies when targeting even distributions, particularly for non-neutral concepts. Our method is specifically designed to address neutral concepts where all classes are factually plausible, as stated on Line 320.
>
> In scenarios where fairness and factual accuracy conflict, our approach prioritizes fairness for neutral cases while avoiding explicit interventions in contexts requiring adherence to factual correctness. This ensures that our method remains applicable and effective within its intended scope.
>
> \
> **Motivation and theoretical discussion**
>
> The motivation for the adaptive mapping network and anchor loss is discussed in detail in Section 3.1, “Is Textual Inversion Effective for De-biasing?” Specifically, we identify the limitations of textual inversion in addressing fairness, such as the imbalanced debiasing performance across different domination groups and the concept drifting issue. These issues motivate the need for a more flexible and concept-preserving mechanism to achieve balanced outputs.
>
> The adaptive mapping network is designed to address these shortcomings by introducing a learnable transformation of text embeddings, enabling context-aware adjustments. The anchor loss further restricts the effect of the inclusive token. Together, these components provide a theoretically grounded and practically motivated framework for mitigating biases in T2I models.
>
> \
> **Specific structure and training process of adaptive mapping network**
>
> The specific structure and training process of the adaptive mapping network are detailed in Section 4.1, “Implementation Details”.
>
> We recognize that the diagrams may lack clarity and will ensure they are revised in the final version to better illustrate the framework and convey our intentions effectively. Additionally, we will further emphasize key implementation details in the text to enhance understanding.
>
> \
> **Quality of generated models**
>
> Regarding the "quality of generated models," we evaluate our generation results using standard metrics such as FID (for image quality) and CLIP score (for text-image alignment), as detailed in Section 4.1 Evaluation Metrics. These metrics provide a quantitative assessment of the quality and consistency of the generated outputs.
>
> As for the "impact of prompts, models, and data volume on the final model," our experiments use well-defined prompts and consistent model settings to ensure fair comparisons. The effect of these factors is controlled to focus on evaluating fairness and quality improvements. If the reviewer could clarify the specific concerns, we would be happy to provide additional explanations or conduct further analysis if needed.
>
> \
> **Experiment setup**
>
> Our setup includes 2.4K images per attribute for $D_{KL}$ calculation, which is notably larger than the 1K images used in ITI-GEN, ensuring a robust evaluation. For complex scene experiments, we focus on demonstrating that the learned adaptive inclusive token is not overfitted to specific sentence structures or image layouts (e.g., close-up human figures). We believe the three examples provided sufficiently prove this point.
>
> Regarding the mention of "limited datasets," we clarify that our experiments are not tied to any specific dataset but rather use a diverse range of prompts to generate images for evaluating the method's effectiveness. If the reviewer could elaborate further on the concerns, we would be happy to address them in more detail.
>
> \
> **Discussion on practical use**
>
> Our method is particularly valuable in real-world T2I applications where users seek to achieve fair and balanced outputs regarding certain target attribute (e.g., gender) without explicitly specifying particular attribute classes (e.g., "female"). This ensures that the generated images maintain diversity while addressing fairness concerns. For example, in scenarios such as content generation, inclusive advertising, or educational tools, our approach enables the generation of unbiased outputs without requiring extensive user intervention.

---

> > ### Comment · Reviewer_SjSS · 2024-11-26
> > **from 3 to 5**
> >
> > Thank you for your feedback. Many of my concerns have been addressed, so I am prepared to increase my score to 5. However, like other reviewers, I still believe that this work has not truly addressed the fairness issues in generative models, including the accuracy of evaluation and the robustness of the method.

---

> > > ### Author Response · Authors · 2024-11-26
> > >
> > > We sincerely thank Reviewer SjSS for constructive feedback and for an increase in score. We are grateful for the acknowledgment that many of the previous concerns have been addressed. We would like to respectfully address the remaining concerns regarding evaluation accuracy, and robustness.
> > >
> > > \
> > > Regarding evaluation accuracy, as mentioned in our previous responses, while CLIP is trained on real-world data and may exhibit minor biases, it is widely accepted in the bias mitigation literature as a robust tool for classifying sensitive attributes and ensuring comparability with prior work. The reliability of our results, shown across multiple experiments, demonstrates that CLIP remains effective in evaluating fairness in generative models.
> > >
> > > \
> > > Furthermore, our method's ability to effectively **mitigate biases across all three sensitive attributes** highlights its robustness. Tables 3 and 4 provide additional evidence of this robustness, demonstrating the method's **capability to address multiple biases simultaneously and handle more complex prompts**. This flexibility, along with its **generalization to unseen concepts**, reflects the method's potential for robust performance in diverse real-world scenarios.
> > >
> > > \
> > > We hope these clarifications address your remaining concerns. We kindly ask you to consider these points when finalizing your assessment, and we deeply appreciate your thoughtful engagement with our submission.

---

### Official Review · Reviewer_QTbU · 2024-11-11

**Soundness:** 3
**Presentation:** 3
**Contribution:** 2
**Rating:** 5
**Confidence:** 4

**Summary:**

The paper focuses on addressing the bias problem in current T2I models. To tackle this, the authors introduce a prompt-tuning approach based on an adaptive mapping network and anchor loss. Experimental results demonstrate its effectiveness in mitigating gender, race, and age biases.

**Strengths:**

1. The paper focuses on addressing the bias problem in current T2I models, particularly gender, race, and age biases, which has important social benefits.

2. The proposed method is simple yet model-agnostic, as stated by the authors, and can be easily adopted in existing approaches to help reduce bias issues.

3. Extensive experiments were conducted to evaluate the effectiveness of the proposed method in mitigating biases.

**Weaknesses:**

1. The biases present in current T2I models are not primarily due to the models themselves but stem from other non-technical issues, such as dataset limitations or unclear prompts. Additionally, for evaluation, the authors use a CLIP zero-shot classifier to classify sensitive attributes. However, since CLIP is likely trained on biased datasets, the classifier may more accurately identify attributes that are highly frequent in its training data, which could, in turn, affect the accuracy of evaluation results.

2. From the quantitative comparison shown in Table 1 and the qualitative results, the proposed method does not show significant improvement over other methods, particularly FM, in addressing different biases. As discussed in the paper, the proposed method is model-agnostic—can it enhance the performance of other methods?

3. What are the experimental settings for Figure 1(a)? More details are needed to illustrate the bias problems present in current T2I models.

4. I am confused about the ablation studies shown in Table 2. What is the setting for without adaptive mapping network? Without AM, how is the anchor loss in Eq. (2) calculated?

**Questions:**

See above weaknesses.

---

> ### Author Response · Authors · 2024-11-22
>
> We thank Reviewer QTbU for recognizing the importance of addressing biases in T2I models and the social benefits of our work. We also appreciate your acknowledgment of the simplicity, model-agnostic nature, and extensive experimental validation of our method. The concerns are addressed as follows:
>
> \
> **Non-technical causes of biases in T2I model**
>
> We appreciate the reviewer's point that biases in T2I models often originate from dataset limitations. Prior works [1, 2] confirm that biases are predominantly inherited and amplified from training data. However, unbiased datasets are impractical to curate in real-world scenarios. Given the increasing prevalence and societal impact of T2I models, it is critical to develop methods that mitigate these biases from a model-centric perspective to ensure fairer and more inclusive outputs.
> Regarding the mention of “unclear prompts” as a source of bias, we respectfully disagree. Bias in T2I models primarily involves the assumption of particular social attributes based on implicit associations (e.g., associating "doctor" with male figures in response to the prompt "a doctor"). Bias mitigation methods directly address these issues by ensuring outputs remain consistent with the intended neutrality of prompts.
>
> [1] The Bias Amplification Paradox in Text-to-Image Generation. NAACL, 2024. \
> [2] Survey of Bias in Text-to-Image Generation: Definition, Evaluation, and Mitigation. ArXiv, 2024.
>
> \
> **Potential inaccuracy caused by biased CLIP model**
>
> We acknowledge the reviewer's concern regarding the potential bias in the CLIP zero-shot classifier due to its training on real-world datasets. To assess this, we conducted an experiment by generating gender-specific figures for stereotypically male- and female-dominated occupations (e.g., "a photo of a female doctor"). Using the neutral prompt "a photo of a doctor," we calculated the CLIP scores for male and female figures, as shown in the table below:
>
> | Types                          | Male Figures | Female Figures |
> |--------------------------------|--------------|----------------|
> | Male-dominated Occupation      | 0.2916       | 0.2780         |
> | Female-dominated Occupation    | 0.2830       | 0.2894         |
>
> The results indicate that while CLIP scores exhibit a slight bias towards more frequent attribute classes, the differences are marginal. Although this limitation is recognized, our use of CLIP follows established practices in the literature [1, 2, 3] to classify sensitive attributes. This alignment ensures consistency and comparability with prior work.
>
> [1] ITI-GEN: Inclusive Text-to-Image Generation. ICCV, 2023. \
> [2] Debiasing Text-to-Image Diffusion Models. ArXiv, 2024. \
> [3] FairRAG: Fair Human Generation via Fair Retrieval Augmentation. CVPR, 2024.
>
> \
> **No significant improvement in Table 1 and qualitative results**
>
> In Table 1, we report KL divergence as the fairness metric. We note that KL divergence inherently has an upper limit (0.6931 for binary cases) and exhibits non-linear degradation (slowing as it approaches zero). This behavior may obscure incremental improvements. Regarding FID and CLIP scores, our primary goal is fairness improvement, not outperforming others in these metrics. The combined results demonstrate that our method achieves fairer outputs without compromising image quality or text-image alignment.
> While our method achieves similar quantitative results to FM, we emphasize FM’s significant degradation in text-image alignment, as illustrated in the third row of Fig. 3(b) and Fig. 4. Such misalignment is critical in T2I tasks, which is also reflected by FM's lower CLIP scores in Table 1.
>
> Qualitatively, our method produces results that are diverse in target attributes while avoiding issues observed in other methods, such as multiple subjects in EI or invisible occupation concepts in FM. These aspects demonstrate that our method presents significant improvements over prior approaches in generating fair and coherent T2I results.
>
> \
> **Enhance the performance of other methods**
>
> Our method is indeed model-agnostic, provided the text encoder can handle the newly learned embeddings. The adaptive mapping network is trained to transform original occupation-related text embeddings into pseudo embeddings, which are then combined with the original embeddings to achieve fairer outputs. As long as the text embeddings remain unaltered by the other methods, our approach can be seamlessly integrated to enhance their performance without compromising compatibility.

---

> ### Author Response · Authors · 2024-11-22
>
> \
> **Experiment setting of Figure 1 (a)**
>
> In Figure 1(a), we evaluate bias in current T2I models using two sets of stereotypically gender-dominated occupations: male-dominated ("doctor," "chief," "farmer," "architect," "software developer") and female-dominated ("ballet dancer," "yoga instructor," "cosmetologist," "fashion designer," "flight attendant"). For each occupation, we generate 100 images using Stable Diffusion 1.5 (SD1.5) with the prompt "A photo of a {occupation}."
> The first-row results demonstrate significant bias in SD1.5: male-dominated occupations predominantly produce male figures, while female-dominated occupations generate mostly female figures. For TI (a fixed inclusive token method), biases are reduced but inconsistently, with more significant reductions for female-dominated occupations, leading to amplified imbalance across sets.
> In contrast, our method employs an adaptive inclusive token, enabling customized adjustments that shift the distribution toward equality rather than disproportionately favoring one gender. This results in more balanced and fair outcomes across both occupation sets.
>
> \
> **Setting of Table 2**
>
> The adaptive mapping network and the anchor loss are independent components. In the "without adaptive mapping network" setting, the Text Transformer (Figure 2, right) directly takes $V_{T_i}$ as input, bypassing the adaptive mapping network. In this configuration, the placeholder $<i>$ embeddings are the only learnable parameters, similar to the textual inversion approach.
> The anchor loss in Eq. (2) is still computed as described, as it depends on the generated embeddings and not on the adaptive mapping network itself. This setup isolates the impact of the adaptive mapping network, allowing us to evaluate its contribution to fairness.

---

> ### Comment · Reviewer_QTbU · 2024-11-26
> **Feedback to the authors' comments**
>
> Thank you for addressing my concerns. However, the main issue regarding accuracy remains unclear. I also agree with other reviews regarding the motivation problem. The authors claimed "bias" observed in current large T2I models may stem from social structures—for example, differing gender ratios in certain occupations. Additionally, providing a clear and specific prompt, such as explicitly including gender, could effectively mitigate the claimed bias in large models. Based on this, the proposed solutions to address this "bias" may offer limited social benefits. In summary, I would like to maintain my current ratings.

---

> ### Author Response · Authors · 2024-11-26
> **Clarification of Misunderstanding Regarding Fairness Definition in the Bias Mitigation Field**
>
> We thank Reviewer QTbU for their thoughtful follow-up and for raising these concerns, as they allow us to further clarify key aspects of our work.
>
> \
> First, regarding the accuracy of the CLIP zero-shot classifier, we acknowledge the potential bias stemming from its training data, as mentioned in our initial response. However, this bias is minor and does not significantly affect its effectiveness for attribute classification in our experiments. The use of CLIP zero-shot classifiers for sensitive attribute classification is a common practice in bias mitigation literature [1,2,3,4,5] due to its robust performance and broad applicability. Additionally, the **consistent and meaningful results across our experiments** demonstrate that the metrics derived from CLIP remain reliable for evaluating fairness in generative models.
>
> \
> Second, we believe there is a misunderstanding regarding fairness in the bias mitigation field. Bias mitigation is not simply about enabling the model to follow explicit prompts better—for example, generating a female doctor when explicitly instructed with “female doctor.” Such cases merely demonstrate the model’s ability to follow a conditioned prompt and do not reflect its inherent fairness. **True fairness lies in how the model responds to neutral prompts where sensitive attributes are left unspecified**, such as “a doctor” or “a yoga instructor.” In such scenarios, models often reflect societal biases present in their training data, leading to skewed outputs (e.g., associating "doctor" with male figures and "yoga instructor" with female figures). Our work, as well as the entire bias mitigation literature, seeks to address this fundamental issue of inherent bias, which is critical to enabling fair and inclusive generative outputs.
>
> \
> Additionally, as pointed out in the ITI-GEN paper, "directly expressing the desired attributes in the prompt often leads to sub-optimal results due to linguistic ambiguity or model misrepresentation." This demonstrates that even when sensitive attributes are explicitly included, the model may fail to produce fair or meaningful outputs due to its inherent limitations in interpreting prompts. This further validates that bias mitigation is a non-trivial challenge requiring solutions beyond explicit prompt specification. Our approach addresses this gap by introducing adaptive inclusive tokens that enable fairer outputs in neutral cases, while also mitigating potential linguistic ambiguities.
>
> \
> Finally, regarding the proposed solutions' social benefits, we respectfully disagree that they are limited. By enabling more inclusive outputs for neutral prompts, our method reduces reliance on explicit attribute specification and empowers broader and fairer usage of T2I models in diverse real-world applications.
>
> \
> We hope this clarifies the points of misunderstanding and demonstrates the importance of our work. We kindly ask the reviewer to reconsider their assessment in light of these explanations, as we believe our method contributes meaningfully to addressing fairness challenges in generative models and provides valuable benefits for inclusive AI applications.
>
> \
> [1] ITI-GEN: Inclusive Text-to-Image Generation. ICCV, 2023. \
> [2] How well can Text-to-Image Generative Models understand Ethical Natural Language Interventions? EMNLP, 2022 \
> [3] Fair Text-to-Image Diffusion via Fair Mapping. arXiv, 2023. \
> [4] Balancing Act: Distribution-Guided Debiasing in Diffusion Models. CVPR, 2024. \
> [5] FairRAG: Fair Human Generation via Fair Retrieval Augmentation. CVPR, 2024.

---

### Author Response · Authors · 2024-12-02
**Responses to all reviewers**

We sincerely thank all reviewers for their thoughtful feedback and for raising important concerns regarding our work. We address below the key issues raised across the reviews, particularly around motivation and robustness.

\
**Motivation**: Several reviewers expressed concerns about whether our work truly addresses fairness in generative models. We believe these concerns are caused by a misunderstanding of the bias mitigation field. Bias mitigation is not about generating accurate results when explicitly conditioned (e.g., "female doctor") but rather ensuring balanced outputs in neutral prompt scenarios (e.g., "a doctor"). Besides, as noted in ITI-GEN [1], relying on explicit conditions in prompts may yield suboptimal results due to linguistic ambiguity or model misrepresentation. Our method addresses these limitations by introducing adaptive inclusive tokens that reduce the reliance on explicit specifications and mitigate inherent biases in generative models. This approach is critical for fairness and inclusiveness, especially given the widespread use of text-to-image systems in real-world applications.

\
**Robustness**: Concerns about robustness were raised in relation to evaluation accuracy, generalizability, and the trade-off with image quality. Regarding evaluation accuracy, we clarify that CLIP zero-shot classifiers, while not perfect, are widely accepted in the bias mitigation literature [1,2,3,4,5] and provide reliable metrics for assessing fairness. Additionally, our method demonstrates generalizability by mitigating multiple biases simultaneously, as shown in Tables 3 and 4. While our method results in a slight compromise in ImageReward scores compared to SD1.5, this trade-off is minor compared to the large degradation observed in Fair Mapping [3]. Furthermore, qualitative results (in Figure 3, 4, 7, 8, 9, 10, 11, 12) clearly show that our method maintains image quality comparable to the original SD models, even for complex prompts.

\
We hope these clarifications address the reviewers' concerns and highlight the novelty, practicality, and impact of our work in addressing fairness challenges in generative models. We sincerely thank the reviewers for their valuable feedback and respectfully ask them to consider raising their scores in consider of the above points.

\
[1] ITI-GEN: Inclusive Text-to-Image Generation. ICCV (Best Paper Candidate), 2023 \
[2] How well can Text-to-Image Generative Models understand Ethical Natural Language Interventions? EMNLP, 2022 \
[3] Fair Text-to-Image Diffusion via Fair Mapping. arXiv, 2023 \
[4] Balancing Act: Distribution-Guided Debiasing in Diffusion Models. CVPR, 2024 \
[5] FairRAG: Fair Human Generation via Fair Retrieval Augmentation. CVPR, 2024

---

### Meta-Review · Area_Chair_cbWN · 2024-12-25

**Metareview:**

This paper introduces AITTI, a method using adaptive inclusive tokens to mitigate biases in text-to-image generation. The approach involves tuning an adaptive mapping network with inclusive samples using an anchor loss to mitigate biases without requiring explicit attribute specification or prior knowledge of the bias distribution.

While the targeted problem is a critical issue in AI ethics and fairness, multiple concerns raised by the reviewers question the motivation behind the proposed approach, its robustness, and its limited advancement beyond existing methods. Considering these factors, I think this paper is not yet ready for publication at ICLR, and thus I recommend rejecting this paper.

**Additional Comments On Reviewer Discussion:**

The reviewers raised several weaknesses in the paper, including concerns about the motivation and scope of bias mitigation, evaluation metrics, and robustness. While the rebuttal successfully addressed some technical misunderstandings and clarified certain issues, the core concerns regarding the motivation behind the proposed method and its robustness remain largely unresolved.

---

### Decision · Program_Chairs · 2025-01-22

Reject